# MEDICONFUSION: CAN YOU TRUST YOUR AI RADIOLOGIST? PROBING THE RELIABILITY OF MULTI-MODAL MEDICAL FOUNDATION MODELS

**Mohammad Shahab Sepehri,[1]  Zalan Fabian,[1]  Maryam Soltanolkotabi,[2]  Mahdi Soltanolkotabi[1]**
[1]Dept. of Electrical and Computer Engineering, University of Southern California
[2]Dept. of Radiology and Imaging Sciences, University of Utah
sepehri@usc.edu

## ABSTRACT

Multimodal Large Language Models (MLLMs) have tremendous potential to improve the accuracy, availability, and cost-effectiveness of healthcare by providing automated solutions or serving as aids to medical professionals. Despite promising first steps in developing medical MLLMs in the past few years, their capabilities and limitations are not well understood. Recently, many benchmark datasets have been proposed that test the general medical knowledge of such models across a variety of medical areas. However, the systematic failure modes and vulnerabilities of such models are severely underexplored with most medical benchmarks failing to expose the shortcomings of existing models in this safety-critical domain. In this paper, we introduce MediConfusion, a challenging medical Visual Question Answering (VQA) benchmark dataset, that probes the failure modes of medical MLLMs from a vision perspective. We reveal that state-of-the-art models are easily confused by image pairs that are otherwise visually dissimilar and clearly distinct for medical experts. Strikingly, all available models (open-source or proprietary) achieve performance below random guessing on MediConfusion, raising serious concerns about the reliability of existing medical MLLMs for healthcare deployment. We also extract common patterns of model failure that may help the design of a new generation of more trustworthy and reliable MLLMs in healthcare. The dataset and evaluation code are available at `https://github.com/AIF4S/MediConfusion`.

## 1 INTRODUCTION

Multimodal Large Language Models (MLLMs) have demonstrated unprecedented capabilities in a variety of multimodal tasks, including image understanding and visual reasoning, autonomous driving (Cui et al., 2024), robotics (Wang et al., 2024a) and embodied AI (Driess et al., 2023). Motivated by this success, a growing body of work (Moor et al., 2023; Li et al., 2024; Lin et al., 2023) explores the potential of MLLMs in medical applications with the hope of paving the way to more accurate, personalized and cost-effective healthcare solutions through modern generative AI.

Even though MLLMs show enormous potential in a wide range of tasks, a swath of challenges have stymied their deployment, including object hallucinations Li et al., relationship hallucinations (Wu et al., 2024a), inaccurate object counting (Jain et al., 2024) and lack of spatial reasoning capabilities (Kamath et al., 2023). These shortcomings are especially worrisome in safety-critical applications, such as healthcare, where reliability is an essential requirement. In fact, recent research efforts on medical MLLMs have revealed weak anatomic knowledge (Nan et al., 2024), concerns on toxicity and patient privacy Xia et al. (2024), highly unreliable disease diagnosis Wu et al. (2023a), and the fact that even a junior doctor far outperforms the most proficient medical MLLMs across a wide spectrum of tasks Wang et al. (2024b). As model failure in the medical domain can lead to serious adverse health effects, it is of utmost importance to understand the performance and limitations of generative AI in the medical context.

A flurry of activity has emerged around probing the performance of medical MLLMs in a multitude of tasks, including visual question answering (VQA) (Ben Abacha et al., 2021), disease classification and report generation (Royer et al., 2024), and modality recognition (Wu et al., 2023b). Even though

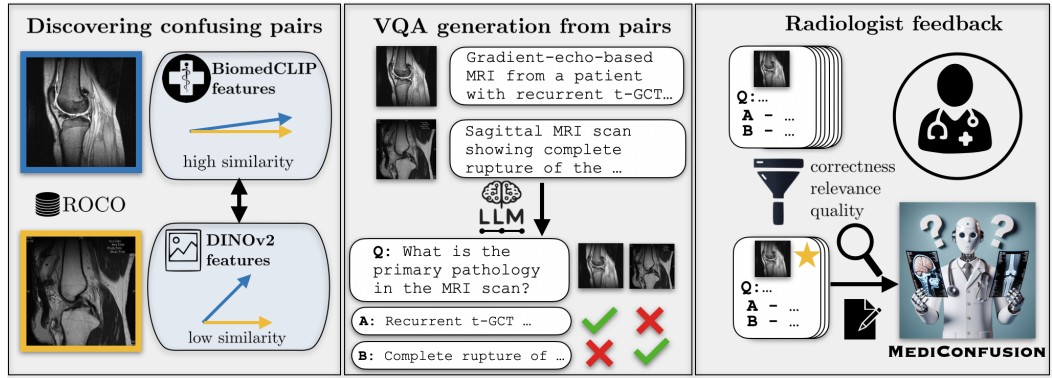

Figure 1: Overview of MediConfusion curation pipeline. First, we extract image pairs from the ROCO radiology dataset that are clearly distinct in the image domain, but may be challenging to differentiate between for multimodal models (left). Next, we use an automated pipeline leveraging LLM prompting to generate VQA from the confusing pairs and their corresponding captions (center). Finally, we incorporate radiologist feedback to filter questions for correctness, relevance and quality, and to revise the questions and answer options for improved medical language and precision (right).

the proposed medical benchmarks offer valuable insights on model performance across a variety of anatomic regions and imaging modalities, they are focused on evaluating the medical knowledge of MLLMs across large evaluation sets, heavily biased towards common or typical scenarios. Therefore, it is unclear how well the measured performance correlates with the actual multimodal medical reasoning capabilities of these models, especially in the face of systematic but perhaps more intricate model failures underrepresented in the dataset. Therefore, developing new evaluation benchmarks that carefully test and probe the capabilities of these systems, expose their vulnerabilities, and facilitate the development of a better understanding of failure modes is vital in healthcare applications.

In this work, we introduce MediConfusion, a challenging benchmark for evaluating the failure modes of medical MLLMs from a vision perspective. We combine novel insights on the image representations of medical MLLMs with the expertise of clinical radiologists to craft a benchmark dataset designed to evaluate the ability of state-of-the-art models to recognize subtle, yet clinically meaningful differences between medical images. Our work reveals that medical MLLMs often confuse image pairs that otherwise appear very different in the image domain. Leveraging this observation, we introduce an automated pipeline to discover such pairs in the ROCO (Pelka et al., 2018) multimodal radiology dataset. Then, in collaboration with radiologists, we curate a VQA benchmark of clinically relevant multiple-choice problems designed to probe the model's ability to distinguish between such images. By design, relying solely on unimodal (language) priors cannot achieve better than random guessing accuracy on our benchmark, and therefore performance on MediConfusion directly correlates with multimodal medical reasoning and image understanding capabilities. Remarkably, we discover that both state-of-the-art medical MLLMs, as well as the most advanced proprietary models, are easily confused by the image pairs, resulting in performance *below random guessing* for most models at the time of writing this paper. What is striking about this poor performance is that for some of the models (i.e. all medical MLLMs we studied) the images and corresponding captions are part of the training data![1] Finally, we leverage our pipeline to categorize failure cases in order to guide future research toward more reliable medical AI solutions.

## 2 THE MEDICONFUSION BENCHMARK

The majority of existing multimodal foundation models leverage CLIP (Radford et al., 2021) to encode the input image (Li et al., 2024; Liu et al., 2024; Moor et al., 2023; Li et al., 2023). CLIP has been pretrained on internet-scale general domain data and, therefore, may not be suitable for the nuanced representation of medical images due to the considerable distribution shift. Thus, variants

---

[1]Given the public nature of the original dataset such images and captions are also likely part of the pre-training of proprietary models such as OpenAI's GPT-4o, Google Deep Mind's Gemini 1.5 Pro, and Anthropic's Claude 3 Opus.

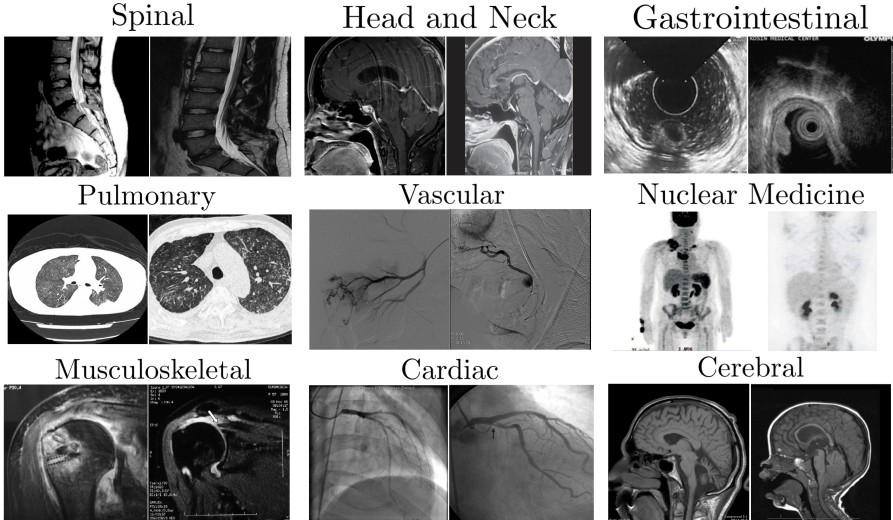

Figure 2: Sample confusing image pairs we have extracted from the ROCO dataset across 9 categories.

trained on large-scale medical image-text datasets have been introduced as image encoders for medical agents, including BiomedCLIP (Zhang et al., 2023b) and PMC-CLIP (Lin et al., 2023). Due to the specialized training data, these models are able to better capture the structure and semantics of medical images. However, surprisingly, we observe that the feature space of even specialized medical encoders is often not rich enough to clearly differentiate between images that are otherwise highly dissimilar. A growing body of recent work Thrush et al. (2022); Yuksekgonul et al. (2023) has shown that the contrastive pretraining objective, shared by common image encoders for MLLMs, can be optimized via shortcuts that lead to fundamental flaws in multimodal understanding. In particular, training consists of aligning image features with their corresponding text features within a batch of data. Thus, if the images are clearly distinct within the batch, the task becomes easy and the model is not encouraged to learn embeddings nuanced enough for more intricate downstream tasks, such as medical reasoning. As a result, MLLMs that leverage such pretrained encoders suffer from impaired image understanding and visual reasoning (Tong et al., 2024a;b), casting serious doubt on the reliability of such models in critical medical diagnosis. Therefore, designing challenging benchmarks that stress-test the visual capabilities of medical MLLM is of utmost importance for gaining a better understanding of the limitations of existing models.

In this work, we introduce the MediConfusion Benchmark, a challenging multiple choice medical visual question answering benchmark designed to probe the reasoning capabilities of medical MLLMs. The overview of our curation pipeline is summarized in Figure 1. First, we extract image pairs that are visually clearly different, but MLLMs will likely confuse them due to their similar features in embedding space. Next, based on the captions corresponding to each of the images in the confusing pairs, we generate a large pool of multiple choice problems via LLM prompting. Finally, each question in the LLM-generated pool is scrutinized and revised by an expert radiologist before being added to MediConfusion. We evaluate a range of state-of-the-art medical and general domain MLLMs and demonstrate that even flagship proprietary models have performance worse than random guessing.

## 2.1 DISCOVERING CONFUSING PAIRS

We find confusing image pairs in ROCO (Pelka et al., 2018), a multimodal dataset of $\approx 80k$ radiology images and their corresponding captions extracted from PMC-OA (Lin et al., 2023) (Figure 1, left). Inspired by Tong et al. (2024b), we seek out pairs with clear visual differences, but high similarity in the feature space of medical vision-language models. This implies that at least one of the images in the pair is compressed ambiguously, and thus, it is likely that relevant visual information is lost in the encoding. In particular, we base our selection criteria on BiomedCLIP's embedding space, as this model has been specifically trained on medical image-text data, and thus it has a more refined feature space for medical images than a general-domain encoder, such as CLIP. Simply put, a radiology image pair that confuses BiomedCLIP, will likely confuse CLIP as well. Moreover, BiomedCLIP has been

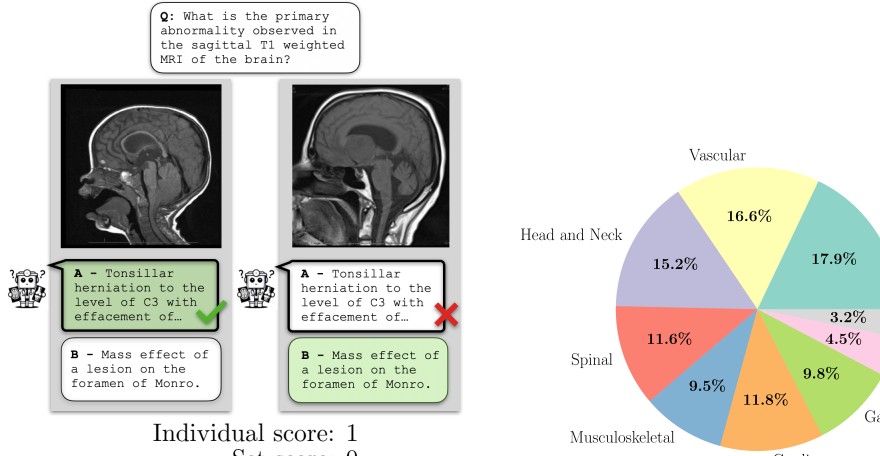

Figure 3: A VQA pair from MediConfusion. A confusing pair shares the same question and answer options, but the correct answer is different for the two (A for the image on the left and B for the image on the right). The model receives a *set score* only if it correctly answers both questions in the confusing pair. *Individual score* is evaluated separately for each image.

Figure 4: Distribution of question categories in MediConfusion. We assign a category to the question based on the category of the corresponding image in the VQA. A single image can belong to multiple categories at the same time.

pretrained on the largest dataset of medical image-caption data among publicly available CLIP-style biomedical vision-language models. We measure visual differences between images in the feature space of DINOv2, a vision-only foundation model with robust image representations that capture visual details. We randomly sample pairs of images and evaluate their similarity in BiomedCLIP ($sim_{med}$) and DINOv2 ($sim_{gen}$) feature spaces. We consider them a confusing pair if $sim_{med} \geq 0.9$ and $sim_{gen} \leq 0.75$ hold at the same time. The gap $|sim_{med} - sim_{gen}|$ can be increased further in order to obtain more difficult pairs, however we find that our setting is already challenging enough for most contemporary models. We depict sample pairs uncovered by our technique in Figure 2.

## 2.2 VQA GENERATION

Given a pool of candidate confusing pairs, we generate multiple choice medical VQA problems that probe the MLLM's ability to effectively differentiate between the images in the pair (Figure 1, center). First, we filter the candidate pool by removing images with short captions ($< 100$ characters) that likely contain insufficient detail about the image. Next, we pass the pair of captions to an LLM (GPT-4) and prompt the model to *generate a question to which the answer is different for the two images* and to provide the two answer options. Thus, we create two VQA problems for each pair that share the same question and answer options, however the correct answer is different for the two images (Figure 3). Therefore, if the medical MLLM is unable to differentiate between the input images, it would only be able to answer at most one of the pair of VQA problems correctly, but not both. As a result, our benchmark by construction cannot be solved to higher than $50\%$ accuracy by relying solely on language prior. In particular, we only credit a *set score* to the model on a particular question pair, if the question has been answered correctly for both images. On the other hand, as a less strict metric an *individual score* is awarded to the model for each correct answer, irrespective of correctness on the other image in the pair. Furthermore, in order to categorize questions in the VQA, we prompt the LLM to assign the most relevant medical area to each of the questions based on the corresponding image's PMC caption. We leverage these categories to break down the performance of existing medical MLLMs across the various categories. We include all prompts used in VQA generation in Appendix A.

## 2.3    DATA FILTERING AND REVISION VIA RADIOLOGIST FEEDBACK

As we generate the questions/answer choices using an LLM, various issues may arise such as factual errors and inconsistencies in quality, format, or language. To ensure the curation of a reliable benchmark dataset, oversight and feedback from a radiology expert are crucial. A radiologist has evaluated each of the automatically generated VQA problems focusing on three aspects.

**Correctness**: the question has to be valid with respect to both of the images, the problems need to be solvable by looking at the individual images alone, and the corresponding answers have to be correct.

**Relevance**: the question has to be relevant to clinical practice or medical research.

**Language**: the problem has to use proper medical terminology and precise language.

Based on these guidelines, the radiologist has assigned a quality score to each question, on a scale $1 - 10$. Higher scores correspond to better problems, and a score of 1 is assigned if correctness is violated in any form (e.g., irrelevant question, incorrect answer). We add a VQA pair to MediConfusion only if the quality score is at least 5 for both individual problems in the pair. Moreover, the expert has verified the medical categories assigned by the LLM to each of the images and revised the question and answer options to improve language quality and precision. This step is crucial in eliminating model artifacts that originate in LLM-generated text inputs.

The resulting benchmark is well-rounded, with questions touching upon 9 areas (see Figure 4): *cerebral*, *spinal*, *cardiac*, *gastrointestinal*, *musculoskeletal*, *vascular*, *pulmonary*, *head and neck*, and *nuclear medicine* with 352 questions in total. The distribution of question categories is depicted in Figure 4.

## 3    EXPERIMENTS

### 3.1    EVALUATION

We evaluate models on MediConfusion based on two notions of accuracy. *Set accuracy* is the portion of correct confusing pairs, where we only consider a pair correct if the model has answered the question correctly for both images in the pair. *Individual accuracy* is the standard notion of accuracy, that is, the portion of correct answers over all questions. An example is depicted in Figure 3. Furthermore, we report *confusion score*, which indicates the portion of pairs where the model has chosen the same answer for both images in the pair, out of all pairs (we exclude pairs where the model generated invalid answers or failed to answer). A high confusion score signifies that the model prediction is overwhelmingly invariant to the specific input image within a pair, and thus, it is confused by the images.

Extracting the knowledge from MLLMs for VQA benchmarking is often challenging due to sensitivity to the specific prompt format and phrasing, strong reliance on language bias and other factors. For instance, instruction tuned models can be directly prompted to answer a multiple choice question with the correct letter option, whereas models without instruction tuning often fail to do so. Therefore, in order to provide a fair comparison, we use a range of evaluation techniques to assess performance.

**Prefix-based score (PS)** – Following Xu et al. (2023), we compute the normalized likelihood of image-question-answer triplets for each answer option, and pick the option with the highest likelihood as the answer. In other words, we select the answer option that the model assigns the highest probability to, given the image and question. To compute the prefix-based score, we concatenate the medical question and the answer sentence directly, stripping the multiple choice question style (e.g., removing "Choose the letter of the correct answer.") and option indicators (e.g., removing "Option A: ...") in order to ensure that models that have not been specifically trained on multiple choice question answering can also consistently provide valid answers.

**Multiple choice prompting (MC)** – We directly prompt the model to answer the multiple choice question with the letter of the correct option. As the output formats may vary (e.g., "A" vs. "The answer is A."), we parse the outputs and attempt to match it to one of the answer options.

**Free-form evaluation (FF)** – We prompt the model to answer the question without providing the answer options or requiring any specific output format. Then, we attempt to match the model output to one of the options using an LLM. In particular, we prompt GPT-4 to score how well the generated

Table 1: Experimental results on MediConfusion. Evaluation techniques: PS - prefix-based scoring, MC - multiple choice prompting, FF - free-form evaluation, GD - greedy decoding evaluation. We underscore the best accuracy for each method across evaluation techniques and report the overall best in **bold**.

| Method | Set acc. (%) | | | | Indiv. acc.(%) | | | | Confusion (%) | | | | Best | |
|---|---|---|---|---|---|---|---|---|---|---|---|---|---|---|
| | MC | GD | FF | PS | MC | GD | FF | PS | MC | GD | FF | PS | Set acc. | Indiv. acc. |
| LLaVA | 8.52 | 9.09 | 1.70 | 1.14 | 50.57 | 51.70 | 15.06 | 49.72 | 85.47 | 85.80 | 76.00 | 97.16 | 9.09 | 51.70 |
| BLIP-2 | 0.57 | 6.82 | 1.70 | 3.98 | 22.16 | 50.28 | 11.65 | 51.42 | 92.19 | 86.93 | 86.67 | 94.89 | 6.82 | 51.42 |
| InstructBLIP | 12.50 | 7.95 | 2.84 | 3.41 | 51.99 | 53.12 | 19.60 | 50.57 | 80.35 | 90.34 | 87.23 | 94.32 | 12.50 | 53.12 |
| DeepSeek-VL2 | 15.91 | 16.48 | 4.55 | 6.25 | 54.26 | 54.26 | 16.19 | 49.43 | 77.19 | 75.57 | 50.0 | 86.36 | 16.48 | 54.26 |
| Molmo | 9.66 | 0.57 | 0.57 | 5.11 | 52.84 | 49.72 | 14.77 | 51.42 | 86.21 | 98.3 | 83.33 | 92.61 | 9.66 | 52.84 |
| LLaVA-Med | 0.00 | 0.00 | 1.14 | 1.14 | 23.58 | 49.72 | 18.75 | 49.72 | 100.00 | 99.43 | 95.92 | 97.16 | 1.14 | 49.72 |
| RadFM | 0.57 | 1.14 | 0.57 | 5.68 | 35.90 | 50.28 | 16.19 | 48.58 | 97.54 | 98.30 | 95.12 | 85.80 | 5.68 | 50.28 |
| Med-Flamingo | 1.14 | 2.27 | 0.57 | 4.55 | 47.73 | 50.00 | 17.05 | 51.99 | 98.75 | 95.45 | 94.89 | 98.30 | 4.55 | 51.99 |
| GPT-4o | 18.75 | - | - | - | 56.25 | - | - | - | 75.00 | - | - | - | 18.75 | 56.25 |
| o1 | 21.59 | - | - | - | 57.95 | - | - | - | 72.99 | - | - | - | 21.59 | 57.95 |
| Claude 3 Opus | 8.52 | - | - | - | 50.85 | - | - | - | 84.09 | - | - | - | 8.52 | 50.85 |
| Gemini 1.5 Pro | 19.89 | - | - | - | 51.14 | - | - | - | 58.52 | - | - | - | 19.89 | 51.14 |
| Gemini 2.0 Flash | 29.55 | - | - | - | 61.93 | - | - | - | 67.05 | - | - | - | **29.55** | **61.93** |
| Random guessing | | | | | | | | | | | | | 25.00 | 50.00 |

output matches each of the options, and we pick the answer option with the highest score. We include the specific evaluation prompt in Appendix A.

**Greedy decoding (GD)–** Similar to multiple choice prompting, we directly prompt the model to answer the problem with the letter of the correct option, then we pick the option with the highest assigned next-token probability. Greedy decoding evaluation is a special case of prefix-based scoring, where the answer options consist of a single letter.

PS and FF evaluations are suitable for models that are not instruction tuned or have not been trained to understand the multiple choice QA format. On the other hand, MC and GD are simpler to evaluate, however these techniques may fail to correctly measure the knowledge of MLLMs unable to understand and follow the multiple choice format. Overall, we represent the performance of each model by their best performance across all evaluation techniques. As we observe, proprietary models can consistently pick an answer option for multiple choice questions; for these models, we only provide MC results. Moreover, output logits necessary for PS and GD evaluation are not available for proprietary models.

We evaluate a representative set of 13 models, 3 of which are medical MLLMs (LLaVA-Med (Li et al., 2024), Med-Flamingo (Moor et al., 2023), RadFM (Wu et al., 2023b)), 5 are flagship proprietary models (GPT-4o, o1, Claude 3 Opus, Gemini 1.5 Pro, Gemini 2.0 Flash) and 5 open-source general-domain MLLMs (LLaVA-7B v1.6 (Liu et al., 2024), BLIP-2 (Li et al., 2023), InstructBLIP (Dai et al., 2023), DeepSeek-VL2(Wu et al., 2024b), Molmo-7B (Deitke et al., 2024)). We set generation parameters according to the corresponding code release and recommended settings and use few-shot prompting for Med-Flamingo (more details in Appendix B).

## 3.2 RESULTS

We summarize the performance of MLLMs on MediConfusion in Table 1. Alarmingly, almost all MLLMs perform below random guessing in terms of set accuracy, corroborating our hypothesis that models struggle to differentiate in fine enough detail between the extracted image pairs necessary for accurate medical reasoning. This observation is further supported by the markedly high (often above 90%) confusion scores, indicating that models tend to select the same answer for both images within a confusing pair. Even RadFM, a model that does not leverage a CLIP-style image encoder, is confused on our benchmark (82.39% confusion score) with performance well below random guessing. As most likely proprietary models leverage visual encoders other than CLIP as well, the overall poor performance and extremely high confusion scores suggest that the exposed vulnerability is more general and not solely rooted in the specific ambiguities of CLIP-style contrastive pretraining.

Table 2: Results by category. We report the best set and individual accuracies (%) for each model across all evaluation techniques.

| Model | Cerebral Set | Indiv. | Vascular Set | Indiv. | Head & Neck Set | Indiv. | Spinal Set | Indiv. | Musculoskel. Set | Indiv. | Cardiac Set | Indiv. | Gastroint. Set | Indiv. | Pulmonary Set | Indiv. | Nuclear Med. Set | Indiv. |
|---|---|---|---|---|---|---|---|---|---|---|---|---|---|---|---|---|---|---|
| LLaVA | 7.59 | 49.37 | 13.70 | 54.79 | 4.48 | 52.24 | 5.88 | 52.94 | 9.52 | 52.38 | 7.69 | 51.92 | **27.91** | **60.47** | 10.00 | 55.00 | 14.29 | 57.14 |
| BLIP2 | 5.06 | 54.43 | 8.22 | 53.42 | 4.48 | 46.27 | 3.92 | 49.02 | 4.76 | 50.00 | 7.69 | 50.00 | 13.95 | 51.16 | 10.00 | 55.00 | 28.57 | 64.29 |
| InstructBLIP | 16.46 | 59.49 | 10.96 | 56.16 | 7.46 | 52.24 | 17.65 | 52.94 | 23.81 | 59.52 | 7.69 | 50.00 | 9.30 | 51.16 | 10.00 | 60.00 | 14.29 | 57.14 |
| DeepSeek-VL2 | 21.52 | 56.96 | 26.03 | 60.27 | **23.88** | 58.21 | 15.69 | 52.94 | 9.52 | 54.76 | 11.54 | 53.85 | 13.95 | 48.84 | 20.00 | 55.00 | 28.57 | 57.14 |
| Molmo | 13.92 | 55.70 | 13.70 | 54.79 | 8.96 | 52.24 | 5.88 | 56.86 | 19.05 | 54.76 | 7.69 | 51.92 | 13.95 | 53.49 | 10.00 | 55.00 | 14.29 | 50.00 |
| LLaVA-Med | 1.27 | 53.16 | 2.74 | 49.32 | 0.00 | 50.75 | 0.00 | 50.98 | 4.76 | 50.00 | 0.00 | 50.00 | 4.65 | 53.49 | 0.00 | 45.00 | 0.00 | 50.00 |
| RadFM | 1.27 | 49.37 | 4.11 | 49.32 | 5.97 | 49.25 | 3.92 | 50.98 | 2.38 | 50.00 | 11.54 | 50.00 | 11.63 | 53.49 | 10.00 | 50.00 | 0.00 | 50.00 |
| Med-Flamingo | 7.59 | 58.23 | 10.95 | 56.16 | 8.96 | 52.24 | 0.00 | 52.94 | 4.76 | 52.38 | 3.85 | 51.92 | 2.33 | 48.84 | 10.00 | 50.00 | 0.00 | 50.00 |
| GPT-4o | 15.19 | 59.49 | 34.25 | 67.12 | 8.96 | 58.21 | 15.69 | 52.94 | 14.29 | 52.38 | 19.23 | 55.77 | 16.28 | 55.81 | **35.00** | **65.00** | 14.29 | 42.86 |
| o1 | **30.38** | **64.56** | 28.77 | 63.01 | 20.90 | **59.70** | 19.61 | 58.82 | 14.29 | 57.14 | 15.38 | 53.85 | 16.28 | 53.49 | 15.00 | 50.00 | **42.86** | **71.43** |
| Claude 3 Opus | 7.59 | 55.70 | 20.55 | 58.90 | 0.00 | 44.78 | 0.00 | 52.94 | 9.52 | 45.24 | 11.54 | 53.85 | 11.63 | 51.16 | 10.00 | 50.00 | 14.29 | 50.00 |
| Gemini 1.5 Pro | 25.32 | 58.23 | 27.40 | 60.27 | 16.42 | 52.24 | 17.65 | 43.14 | 26.19 | 47.62 | 7.69 | 48.08 | 23.26 | 44.19 | 5.00 | 50.00 | 28.57 | 57.14 |
| Gemini 2.0 Flash | 29.11 | **64.56** | **39.73** | **68.49** | 19.40 | 58.21 | **23.53** | **58.86** | **40.48** | **66.67** | **26.92** | **61.54** | 25.58 | **60.47** | 30.00 | 50.00 | **42.86** | **71.43** |

Gemini models are interesting outliers: even though they are the least confused on the dataset, they struggle tackling MediConfusion as well, as Gemini 2.0 barely surpasses random guessing by about 3%. This may suggest that the model's visual representations are rich enough to meaningfully distinguish between images; however, the medical knowledge or necessary reasoning skills are lacking to correctly answer the questions.

Furthermore, perhaps surprisingly, medical MLLMs did not outperform other methods, indicating that the shortcomings cannot be addressed exclusively by domain-specific training. These results are especially surprising, as the image-caption pairs used to generate MediConfusion are part of PMC-OA, which is included in the pre-training set of all medical MLLMs in our experiments. We also note that given the public nature of PMC-OA these image-caption pairs are likely included in the training set of proprietary models as well. Finally, we see some performance gap between open-source and proprietary models, with Gemini 2.0 achieving the highest individual accuracy of 61.93%.

We further break down the results based on the category of the question in order to identify if specific areas have been more/less challenging to the models. We summarize our findings in Table 2. Even though the overall results across all categories are close to random guessing performance, proprietary models demonstrate slightly better accuracies on questions related to cerebral, vascular, and nuclear medicine images. In particular, Gemini 2.0 achieves 42.86% and 71.43% set and individual accuracy correspondingly on nuclear medicine, an overall best across all models and categories

## 4 DISCUSSION

### 4.1 IDENTIFYING PATTERNS IN CONFUSING PAIRS

Our experiments have demonstrated that state-of-the-art MLLMs are easily confused by radiology image pairs that exhibit major differences obvious to human experts. The first step towards improving the reliability of such models is to identify and categorize common cases where medical MLLMs tend to break down. We leverage an expert-in-the-loop pipeline to extract failure modes from MediConfusion via a combination of LLM prompting and radiologist supervision. In particular, we pass the VQA problems from MediConfusion to GPT-4, where we replace the images with their corresponding captions from ROCO. We prompt the model to summarize the key differences between images in a pair that the questions are designed to test (details in Appendix A). The LLM identifies patterns in the extracted differences and distills them into a set of categories that the radiologist corrects and refines based on the dataset. As a result, we identify the following common patterns that have confused the models:

- **Pattern 1: Normal/variant anatomy vs. pathology–** Models often struggle to differentiate between normal/variant anatomy and pathological structures. For instance, the model often confuses malalignment with normal alignment (e.g., atlantoaxial dislocation vs. normal atlantoaxial interval) or differentiating pituitary region masses (suprasellar vs. parasellar vs. intrasellar) or various anatomical regions of the spine (cervical vs. thoracic vs. lumbar).

- **Pattern 2: Lesion signal characteristics–** Models fail to correctly identify regions of high signal intensity and their significance, particularly on T2-weighted sequences. This failure is especially of clinical significance in differentiating solid vs. cystic entities.

- **Pattern 3: Vascular conditions–** Identifying aneurysms and differentiating them from normal vascular structures or other abnormalities like vascular malformations seems to be challenging for MLLMs. Furthermore, there is often confusion between total occlusions and partial stenosis in coronary arteries.

- **Pattern 4: Medical devices–** Models often fail to detect the presence of stents and have difficulties distinguishing between various types of stents. Identifying the presence or absence of guidewires in images of interventional procedures tends to also be challenging for MLLMs.

Most of the above shortcomings can be, to some degree, traced back to known, common failure modes (Tong et al., 2024b) of visual reasoning in MLLMs in the general domain.

**Detecting presence (or absence) of specific features:** Correct reasoning over medical VQA problems strongly relies on detecting the presence (or absence) of particular features or objects relevant to the question. MLLMs are known to suffer from object hallucinations (Li et al.) rooted in parts in flawed image encoding, statistical biases and strong reliance on language priors (Leng et al., 2024). We can see this specific weakness reflected in Patterns 3 and 4 directly.

**Understanding state and condition:** In medical VQA, it is crucially important for the model to understand the difference between "normal" and "abnormal" structures. MLLMs have difficulties identifying the state and condition of objects in the general domain, such as whether the ground is wet or if a flag is blowing in the air (Tong et al., 2024b). These challenges may be amplified in the more nuanced medical setting, which we observe in Patterns 1 and 3 especially.

**Positional and relational context:** Answering medical VQA problems often necessitate a careful understanding of the spatial relationships of various anatomical features and their specific location. Recent research has uncovered serious limitations in the spatial reasoning capabilities of MLLMs (Kamath et al., 2023), some even failing to distinguish left from right. This pervasive weakness in spatial reasoning may translate to failures in medical VQA seen in Pattern 1.

**Color and appearance:** Recent work has shown that MLLMs can confuse colors and their intensity (bright/dark) (Tong et al., 2024b), which may cause challenges in identifying signal characteristics in radiology images (high/low intensity) reflected in Pattern 2.

## 4.2 VISUAL PROMPTS IN MEDICONFUSION

Free-form visual prompts are intuitive annotations in the input image, such as a red bounding box or an arrow, aimed at highlighting a specific point or area within the image. It is natural to ask whether well-placed visual prompts in medical images, annotated by a doctor, can potentially guide the attention of MLLMs to important areas in the image and thus help provide accurate answers. Such a capability would greatly facilitate human-machine collaboration in healthcare and provide more reliable AI-assisted diagnosis. In the general domain, research has shown that MLLMs typically are unable to efficiently interpret visual prompts without incorporating such task specifically into the training procedure (Cai et al., 2024).

We find that some images in MediConfusion include such visual prompts, typically in the form of arrows pointing at the abnormality, and in a specific case the correct answer is written in the image along with the prompt (Figure 5). We observe that only proprietary models, as well as LLaVA v1.6 and BLIP-2, have been able to provide consistently correct answers for this particular image, and none of the medical MLLMs. We hypothesize that the success of proprietary models and LLaVA v1.6 can be attributed to their OCR (optical character recognition) capabilities, which is missing from medical MLLMs. In examples where only the visual prompt (e.g., an arrow pointing at the abnormality/region of interest) is included we don't observe a similar trend. We believe that understanding and improving the visual prompting capabilities of medical MLLMs is a promising direction for future research.

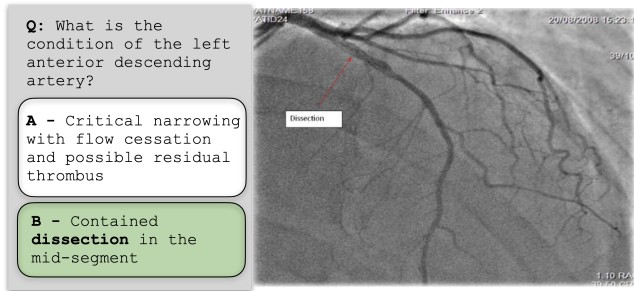

Figure 5: Sample VQA from MediConfusion where the solution is directly provided in the image in the form of text and visual prompts (arrows). Medical MLLMs not trained for OCR have been unable to leverage the hint.

## 5 RELATED WORK

**Multimodal Large Language Models –** Large Language Models (LLMs) such as InstructGPT (Ouyang et al., 2022) and LLaMA (Touvron et al., 2023) have emerged as powerful models capable of performing complex tasks rooted in natural language, including text summarization, coding, and question-answering. LLMs are pretrained on massive text corpora and can be efficiently adapted to downstream tasks. Beyond textual inputs, the LLM pretraining framework has been extended to further modalities, such as images, resulting in multimodal large language models that demonstrate strong visual understanding and reasoning capabilities. LLaVA (Liu et al., 2024) interleaves image representations with the input text of a pretrained LLaMA model and fine-tunes on visual instruction-following data. Flamingo (Alayrac et al., 2022) injects visual information into a frozen LLM via cross-attention. BLIP (Li et al., 2022; 2023) proposes the Q-Former architecture for connecting pretrained vision features to an LLM. More recently, GPT-4o has been trained from scratch on mixed multimodal inputs directly.

Beyond the general domain, MLLMs are especially promising in automating costly medical tasks, such as analyzing radiology images, generating medical reports or acting as medical conversational agents to provide healthcare advice. There has been substantial research recently to develop medical MLLMs, most often by adapting popular general domain architectures to medical data. Med-Flamingo (Moor et al., 2023) pretrains Flamingo on interleaved image-text medical data sourced from publications and textbooks, unlocking few-shot medical VQA capabilities. Authors of LLaVA-Med (Li et al., 2024) focus on rapid adaptation to the medical domain by fine-tuning LLaVA on filtered image-text pairs from PMC-15M (Zhang et al., 2023a). Authors in Zhang et al. (2023c) generate a large-scale medical VQA dataset from PMC-OA (Lin et al., 2023) which is subsequently used to train a medical MLLM. Moreover, authors in Wu et al. (2023b) propose a multimodal foundation model, RadFM, for radiology, aligning natural language with 2D and 3D radiology images.

**Encoding visual information in MLLMs –** The prevailing approach to incorporate visual information in MLLM training leverages contrastive language-image pretrained models as frozen image encoders. CLIP (Radford et al., 2021), and its variants (Cherti et al., 2023), are trained on internet-scale paired image-text data, and thus its representations are readily aligned with natural language, and can be effectively combined with language models. The frozen representations are then adapted to the feature space of the language model using MLP heads (Liu et al., 2024), Q-Former (Li et al., 2022), cross-attention (Alayrac et al., 2022) or other mechanisms. The image encoder acts as the "eye" of the MLLM as it directly determines what visual information enter the model. In fact, imperfect compression of relevant visual information is a dominant issue with contemporary MLLMs, resulting in object hallucinations (Li et al.; Gunjal et al., 2024), fundamental errors in spatial reasoning (Kamath et al., 2023), and inability to understand inter-object relationships (Wu et al., 2024a).

As the distribution of general 'internet data' and medical image-text data is markedly different, CLIP may be unable to capture the intricate structure of medical images with fidelity sufficient for reliable performance. Researchers have proposed CLIP-like models pretrained on large-scale medical data better suited as image encoders for medical MLLMs. LLaVA-Med leverages BiomedCLIP (Zhang et al., 2023b), a foundation model designed for biomedical image-text processing that has been pretrained on PMC-15M. MedVInT uses PMC-CLIP (Lin et al., 2023), a CLIP-style model pretrained on PMC-OA with $1.6M$ medical image-caption pairs. The limitations of image encoders in medical

MLLMs have attracted less attention than in the general domain, which is especially troubling due to the safety-critical nature of healthcare applications. Thus, the lack of in-depth understanding of the shortcomings and possible failure modes of the image encoder in the medical MLLM pipeline is an exceedingly pressing concern.

**Medical VQA benchmarks –** With the recent rapid advances in developing medical MLLMs, there has been substantial effort in quantifying their performance in a wide range of tasks and areas within the medical domain. VQA-Rad (Lau et al., 2018), SLAKE (Liu et al., 2021), Path-VQA (He et al., 2020) and VQA-Med (Ben Abacha et al., 2021) are widely-used to benchmark the performance of MLLMs in medical VQA. Due to their small size and limited scope, there has been a push for more comprehensive and diverse evaluation datasets. OmniMedVQA (Hu et al., 2024) introduces the largest medical VQA dataset to date, encompassing 12 data modalities and 20 anatomical regions with a total of more than $100k$ images. Authors of Asclepius (Wang et al., 2024b) focus on eliminating data leakage present in other benchmarks and providing human evaluations. GMAI-MMBench (Chen et al., 2024) incorporates problems probing the performance of MLLMs at various perceptual granularities, and targets a well-categorized data structure for ease of preparing customized evaluations. Other benchmarks extend the evaluation task beyond VQA in order to provide a more comprehensive view of model performance. MultiMedEval (Royer et al., 2024) builds a uniform and fair benchmarking framework for multiple tasks including report generation and classification. Micro-Bench Lozano et al. (2025) evaluates the visual understanding capabilities of MLLMs across diverse microscopy modalities. RadBench (Wu et al., 2023b) focuses on radiology with associated tasks such as modality recognition and disease diagnosis. Authors of CARES (Xia et al., 2024) aim to provide a more holistic view of model performance by focusing on aspects such as fairness, privacy and safety of MLLMs as well as factual correctness.

All of these datasets are aimed at probing the medical knowledge of MLLMs and quantifying their average performance on a wide variety of tasks, modalities and anatomic regions. However, none of these benchmarks are specifically designed to probe the reliability, fundamental limitations and failure modes in the medical domain, all critical aspects in healthcare applications. Perhaps the closest work in spirit to ours is RadVUQA (Nan et al., 2024), where authors call attention to the critical deficiencies of existing medical MLLMs, revealing a large gap between state-of-the-art MLLMs and clinicians. Their dataset focuses on more fundamental visual question answering and understanding, such as spatial reasoning, anatomic understanding and quantitative reasoning on medical images. However, we go a step further and design a benchmark that stress-tests the visual capabilities of MLLMs by curating questions expected to be challenging for their image processing pipeline.

Related to our work, Tong et al. (2024b) has investigated the failure modes of MLLMs originating in ambiguous vision encoding in the general domain. Their study is based on finding CLIP-blind pairs, images that have high similarity in CLIP embedding space, but otherwise have dissimilar low-level image features. However, their methodology is not directly applicable in the medical domain for two reasons. First, CLIP has been pretrained on general domain data and thus it is unable to capture the intricate structure of medical images. Second, their methodology relies on human annotators to describe the difference between a large number of image pairs, which is prohibitively costly in our scenario, as only radiologists are qualified to provide such annotations in the medical setting.

## 6 CONCLUSION

In this paper, we introduce MediConfusion, a challenging medical VQA benchmark designed to probe the limitations of multimodal reasoning in medical MLLMs. In particular, we discover radiology image pairs that, due to ambiguities originating in their multimodal embedding spaces, confuse contemporary models despite being dissimilar in the image domain. We leverage an automated pipeline along with the expertise of radiologists to create a dataset of VQA problems that tests the ability of MLLMs to effectively distinguish and answer clinically relevant questions about such confusing pairs. Our benchmark, by construction, cannot be solved by leveraging unimodal priors, and thus, it directly probes multimodal capabilities. We find that most existing models achieve performance no better than random guessing on MediConfusion, as models tend to select the same answer option for both images in the pair, raising serious concerns about the reliability of existing MLLMs in a medical setting. In order to guide future research in addressing the limitations of current MLLMs, we identify common failure patterns where models often break and relate them to known limitations in the general domain. We hope that our work sparks further research efforts to improve the reliability of AI for healthcare applications.

ACKNOWLEDGEMENTS

We would like to thank Microsoft for an Accelerating Foundation Models Research grant that provided the OpenAI credits enabling this work. This research is also in part supported by AWS credits through an Amazon Faculty research award and a NAIRR Pilot award. M. Soltanolkotabi is also supported by the Packard Fellowship in Science and Engineering, a Sloan Research Fellowship in Mathematics, an NSF-CAREER under award #1846369, DARPA FastNICS program, and NSF-CIF awards #1813877 and #2008443. and NIH DP2LM014564-01.

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

# APPENDIX

## A PROMPTS FOR DATASET CURATION

In this section, we provide the prompts used to interact with GPT-4o for dataset generation and MLLM evaluation. Wherever we use **TEXT**, we mean that TEXT is a description or variable that is image/pair specific.

### A.1 QUESTION GENERATION

We use the following prompt to generate a question for a single confusing pair. We describe the output format as detailed as possible to be able to process the answers with little human interaction.

**System message:** You are a helpful assistant expert in the medical domain.
**Prompt:** Here is the description of two medical images that I can see:
Image1: **CAPTION OF IMAGE 1**
Image2: **CAPTION OF IMAGE 2**
Your task is to create multiple-choice questions. Follow the rules below.
1. The question should be about a property that is clearly visible in the images.
2. It should be possible to answer the question by only looking at the images.
3. Pretend that you can only see one image. You are not allowed to refer to 'Image1', 'Image2' or 'images'. You should also not create questions that require comparing the two images.
4. There should be exactly two answer options. You have to come up with a question for which the answer is different for the two images.
5. The answer options should be clearly different.
Please provide the question and the two answer options in the following format:
Question: <YOUR QUESTION>
Option1: <ANSWER 1>
Option2: <ANSWER 2>
Also, please provide the correct answers to the question for the images corresponding to our captions in the following format:
Image1: <ANSWER>
Image2: <ANSWER>
Aim for simple, efficient and concise questions to best test someone's knowledge and understanding of the underlying concepts.

### A.2 CATEGORIZING IMAGES

To categorize the images, we first show GPT-4o captions of several (here we use 100) images and ask it to separate them into different categories. Afterward, for each image, we ask GPT-4o to pick one of the categories for that image based on its caption.
We used the following prompt to extract categories:

**System message:** You are a helpful assistant expert in the medical domain.
**Prompt:** I have several medical images related to radiology that each one has a corresponding caption. I have listed the captions below. Can you go through the captions and categorize them?

```
Please focus on the general categories.
Caption 0:  **IMAGE 0 CAPTION**
Caption 1:  **IMAGE 1 CAPTION**
...
Caption 99:  **IMAGE 99 CAPTION**
```

Using this prompt, we find 9 categories: Cerebral, Spinal, Cardiac, Gastrointestinal, Musculoskeletal, Vascular, Pulmonary, Head and Neck, Breast, and Other.
We used the following prompt to assign categories:

**System message:**  You are a helpful assistant expert in the medical domain.
**Prompt:** Caption:  **IMAGE CAPTION**
The above caption describes a radiology image.  To which of the following categories does this image belong?  You should only name the category, and you do not need to specify your reasoning.
Categories:  Cerebral, Spinal, Cardiac, Gastrointestinal, Musculoskeletal, Vascular, Pulmonary, Head and Neck, Breast, Other"

The final set of categories in our dataset are somewhat different, as we incorporated feedback from the radiologist to revise the automatically generated categories.

### A.3    FINDING FAILURE MODES

To find common failure modes that our dataset probes, we use the questions and captions of 100 pairs in the following prompt to send to GPT-4o:

**System message:**  You are a helpful assistant expert in the medical domain.
**Prompt:**   I am analyzing an image embedding model.  I have several image pairs that each one has a corresponding two choice question. I know that the embedding model confuses the images about the corresponding question.  Can you go through the questions, options, and image descriptions, trying to figure out some general patterns that the embedding model struggles with?  Please focus on the visual features and generalize patterns that are important to vision models.
Pair 0
First image description:  **IMAGE 1 CAPTION**
Second image description:  **IMAGE 2 CAPTION**
Confusing multiple choice question:  **QUESTION**
Pair 1
First image description:  **IMAGE 1 CAPTION**
Second image description:  **IMAGE 2 CAPTION**
Confusing multiple choice question:  **QUESTION**
...
Pair 99
First image description:  **IMAGE 1 CAPTION**
Second image description:  **IMAGE 2 CAPTION**
Confusing multiple choice question:  **QUESTION**

## A.4 FREE FORM EVALUATION

For the free-form (FF) GPT-4o evaluation, we pass the MLLM's answers with the following prompt to GPT-4o to obtain two scores, one for each answer option.

**System message:** You are a helpful and precise assistant for checking the quality of the answer.
**Prompt:** [Question]
**QUESTION**
[Answer A]
**OPTION A**

[Answer B]
**OPTION B**

[Assistant]
**RESPONSE**

[End of Assistant]

[System]
We would like to request your feedback on the performance of an AI
assistant in response to the user question displayed above.  The
user asks the question on observing an image.  We have provided
two possible answers, [Answer A] and [Answer B] to the question.
Your job is to evaluate how close the AI assistant's answer is
to each of the answers.  You don't have to decide whether the
answers are correct or not.  Each answer should receive an overall
score on a scale of 1 to 10, where a higher score indicates the
AI assistant's answer is closer to the specific answer.  After
providing the scores, concisely provide your explanation for
the given scores.  Remember, you don't need to comment on the
correctness of the answers.  Please provide your answer in the
following format:
A: <SCORE>
B: <SCORE>
Your explanation:  <EXPLANATION>

These scores are the similarities of the MLLM's answer to the different answer options. If the gap between the higher and lower score is at least $k = 3$, which we call the score gap threshold, we assign the option with the higher score as the MLLM's output. Otherwise, we mark the answer as invalid.

## B MODEL DETAILS

In this section, we provide details on the versions and hyperparameters of MLLMs that we use. It should be noted that for the multiple choice (MC) evaluation mode, we set temperature to $0$, as we only expect a single letter option to be generated.

### B.1 MEDFLAMINGO FEW-SHOT PROMPTING

In order for MedFlamingo to produce valid responses, we need to use few-shot prompting. Here, we show three questions and answers from PMC-VQA benchmarks Zhang et al. (2023c). The following is the prompt we used for MC evaluation:

| MLLM | Version/LLM | Temperature | Beams | Top p |
|------|-------------|-------------|-------|-------|
| LLaVA | v1.6/Mistral 7B | 0.2 | 1 | - |
| BLIP-2 | Opt 2.7B | 1 | 5 | 0.9 |
| InstructBLIP | Vicuna 7B | 1 | 5 | 0.9 |
| DeepSeek | VL2 | 0.7 | - | - |
| Molmo | 7B | 0.7 | 1 | - |
| LLaVA-Med | v1.5/Mistral 7B | 0.2 | 1 | - |
| RadFM | - | - | - | - |
| Med-Flamingo | - | 1 | 5 | 0.9 |
| GPT | 4o (release 20240513) | 0.7 | - | - |
| o1 | (release 20241217) | 0.7 | - | - |
| Claude | 3 Opus | 0.2 | - | - |
| Gemini | 1.5 Pro | 0.2 | - | - |
| Gemini | 2.0 Flash | 0.2 | - | - |

Table 3: MLLM details

**Prompt:**         You are a helpful medical assistant.  You are being
provided with images, a two choice question about each image
and an answer.  Follow the examples and answer the last question.
<image>Question:  What radiological technique was used to confirm
the diagnosis?
A: CT Scan
B: Mammography
Answer:  B: Mammography<|endofchunk|><image>Question:  What did
the CT scan show?
A: Cerebral edema
B: Intracranial hemorrhage
Answer:  A: Cerebral edema|endofchunk|><image>Question:  What is
the purpose of the asterisk shown in the figure?
A: To indicate the formation of lobes around the contracting
nucleus.
B: To indicate the normal lentoid shape of hypocotyl nuclei.
Answer:  B: To indicate the normal lentoid shape of hypocotyl
nuclei.<|endofchunk|><image>
Question:  **QUESTION**
A: **OPTION A**
B: **OPTION B**
Answer:

The following is the prompt we used for FF evaluation:

**Prompt:**            You are a helpful medical assistant.  You are
being provided with images, a question about each image
and an answer.  Follow the examples and answer the
last question.  <image>Question:  What radiological
technique was used to confirm the diagnosis?  Answer:
Mammography<|endofchunk|><image>Question:  What did the CT scan
show?  Answer:  Cerebral edema|endofchunk|><image>Question:
What is the purpose of the asterisk shown in the figure?

```
Answer:  To indicate the normal lentoid shape of hypocotyl
nuclei.<|endofchunk|><image>Question:  **QUESTION** Answer:
```

The following is the prompt we used for GD evaluation:

```
Prompt:         You are a helpful medical assistant.  You are being
provided with images, a two choice question about each image
and an answer.  Follow the examples and answer the last question.
<image>Question:  What radiological technique was used to confirm
the diagnosis?
A: CT Scan
B: Mammography
Answer:  B: Mammography<|endofchunk|><image>Question:  What did
the CT scan show?
A: Cerebral edema
B: Intracranial hemorrhage
Answer:  A: Cerebral edema|endofchunk|><image>Question:  What is
the purpose of the asterisk shown in the figure?
A: To indicate the formation of lobes around the contracting
nucleus.
B: To indicate the normal lentoid shape of hypocotyl nuclei.
Answer:  B: To indicate the normal lentoid shape of hypocotyl
nuclei.<|endofchunk|><image>
Question:  **QUESTION**
A: **OPTION A**
B: **OPTION B**
Answer:
```

The following is the prompt we used for PS evaluation:

```
Prompt:              You are a helpful medical assistant.  You are
being provided with images, a question about each image
and an answer.  Follow the examples and answer the
last question.  <image>Question:  What radiological
technique was used to confirm the diagnosis?  Answer:
Mammography<|endofchunk|><image>Question:  What did the CT scan
show?  Answer:  Cerebral edema<|endofchunk|><image>Question:
What is the purpose of the asterisk shown in the figure?
Answer:  To indicate the normal lentoid shape of hypocotyl
nuclei.<|endofchunk|><image>Question:  **QUESTION** Answer:
**ANSWER**
```

## C  ABLATION ON PROMPTING

In this section, we perform an ablation study on the effect of prompt variations on model performance. We test GPT-4o, InstructBLIP, and LLaVA-Med with MC evaluation. We manually create 10 prompt variations different from the original, shown in Table 4. We do not rely on LLMs for rephrasing the original prompt to avoid biases and artifacts originating in LLM-generated text. To capture variability in accuracy both due to prompting and stochasticity, we sample 10 answers for each model, and for each prompt. Table 5 shows our results.

First, we observe that the performance of GPT-4o is robust to the prompt format, but the performance is still below random guessing. InstructBLIP is more sensitive to the particular input prompt, as demonstrated by significantly reduced performance on some prompts. Lastly, LLaVA-Med struggles

with the MC evaluation, as it has not been specifically trained to follow the multiple-choice format, and prompt engineering does not fix this issue.

Table 4: List of prompts we use to evaluate sensitivity of model performance to prompt variations. The second column shows the high-level rationale for the variation, and the specific prompt is included in the last column.

| # | Change | Prompt |
|---|--------|--------|
| 0 | Original | Based on the image, choose the correct option for the following question.
Question:
A:
B:
Answer with the option's letter from the given choices directly. Your answer should be just one letter.
Answer: |
| 1 | Shorten | Answer the following question about the image.
Question:
A:
B:
Answer with the option's letter directly.
Answer: |
| 2 | Move answer options to the end | Answer the following question about the image with the correct option's letter directly.
Question:
A:
B:
Answer: |
| 3 | Instruct the model for extra care and attention | Carefully and skillfully review the image and answer the following question.
Question:
A:
B:
Answer with the option's letter directly.
Answer: |
| 4 | Define AI expert role | You are an expert radiologist AI with deep knowledge in radiology image analysis. Based on the image, choose the correct option for the following question.
Question:
A:
B:
Answer with the option's letter directly.
Answer: |
| 5 | Rephrase the instruction to answer with a single letter | Based on the image, choose the correct option for the following question.
Question:
A:
B:
Simply respond with 'A' or 'B' based on your answer.
Answer: |
| 6 | Ask the model nicely | Based on the image, please choose the correct option for the following question.
Question:
A:
B:
Please answer with the option's letter from the given choices directly.
Answer: |
| 7 | Remove new lines from formatting | Based on the image, choose the correct option for the following question. Question: Option A: Option B: Answer with the option's letter from the given choices directly. Your answer should be just one letter. Answer: |
| 8 | Give more explanation | You are given a radiology image, based on which you will need to carefully answer a medical question related to the image. You will be given two answer options: A and B from which you have to choose the right answer. The correct answer can always be determined just by looking at the radiology image itself. Take a close look at the image, think carefully, and answer the following question.
Question:
A:
B:
Answer with the option's letter from the given choices directly. Your answer should be just one letter.
Answer: |
| 9 | No instructions | Question:
A:
B:
Correct answer option: |
| 10 | Emphasize importance | You are given a radiology image, and your critical task is to take a close look and choose the correct option for the following question. It is extremely important to answer the following question correctly.
Question:
A:
B:
Answer with the option's letter from the given choices directly. Your answer should be just one letter.
Answer: |

Table 5: Results of prompt variations. The table shows the mean and standard deviation of set accuracy (in %) across 10 random samples for each prompt. The last column aggregates the results across all prompts.

| Model | Prompt 0 | Prompt 1 | Prompt 2 | Prompt 3 | Prompt 4 | Prompt 5 | Prompt 6 | Prompt 7 | Prompt 8 | Prompt 9 | Prompt 10 | All Prompts |
|---|---|---|---|---|---|---|---|---|---|---|---|---|
| GPT-4o | $19.32 \pm 1.53$ | $19.43 \pm 0.66$ | $18.01 \pm 0.95$ | $17.33 \pm 0.96$ | $18.30 \pm 0.80$ | $19.66 \pm 0.81$ | $19.43 \pm 1.62$ | $20.40 \pm 1.12$ | $19.38 \pm 1.15$ | $19.43 \pm 1.50$ | $19.66 \pm 1.11$ | $19.12 \pm 0.04$ |
| InstructBLIP | $11.65 \pm 1.42$ | $5.57 \pm 0.91$ | $3.75 \pm 0.77$ | $4.38 \pm 1.48$ | $2.67 \pm 1.19$ | $1.31 \pm 0.57$ | $4.26 \pm 0.99$ | $4.43 \pm 1.86$ | $1.42 \pm 0.77$ | $12.05 \pm 3.15$ | $1.25 \pm 0.66$ | $4.79 \pm 0.12$ |
| LLaVA-Med | $0.00 \pm 0.00$ | $0.00 \pm 0.00$ | $0.00 \pm 0.00$ | $0.00 \pm 0.00$ | $0.00 \pm 0.00$ | $0.00 \pm 0.00$ | $0.00 \pm 0.00$ | $0.00 \pm 0.00$ | $0.00 \pm 0.00$ | $0.00 \pm 0.00$ | $0.00 \pm 0.00$ | $0.00 \pm 0.00$ |

# D  FINE-TUNING ON MEDICONFUSION

In this section, we investigate the effect of fine-tuning medical MLLMs on MediConfusion. Fine-tuning on the specific benchmark typically helps the model to better understand the format of the VQA and can greatly boost the performance of the model on the particular benchmark.

We create train and test splits from MediConfusion by randomly sampling 50% of confusing pairs from each category to train on, and hold out the other half for evaluation. We ensure that the two images from a confusing pair always belong to the same split in order to avoid leakage. As each pair may belong to multiple categories, the above process results in two splits of slightly different size (84 pairs in train and 92 pairs in test), but with category distribution close to the full dataset. We following common procedure to fine-tune LLaVA-Med on the train split, involving fine-tuning the language model and the multimodal adapter, keeping the image encoder frozen. We perform a search over learning rates within $\left[10^{-6}, 5 \cdot 10^{-6}, 10^{-5}, 10^{-4}, 2 \cdot 10^{-4}\right]$ and select $5 \cdot 10^{-6}$. We keep other fine-tuning hyperparameters at their recommended values. We perform full fine-tuning, as opposed to parameter-efficient fine-tuning approaches, and train for up to 1000 epochs. Train and test (set) accuracy across epochs are depicted in Figure 6.

Interestingly, we observe that the model struggles to achieve 100% set accuracy on the training set, even after a very high number of training epochs (typical fine-tuning is within $1 - 50$ epochs), further supporting our hypothesis that the model is unable to differentiate between image pairs due to fundamental ambiguity in their embeddings. In particular, the vision features appear so similar to the model that even fine-tuning the whole language model is insufficient. This experiment suggests that MediConfusion can only be solved by improving the vision encoder directly.

Furthermore, we find that fine-tuning helps LLaVA-Med learn the multiple-choice format (set accuracy on the test split increased from 0% to around 20%), however its performance is still well below random guessing as the knowledge learned on the training set does not generalize to the test set.

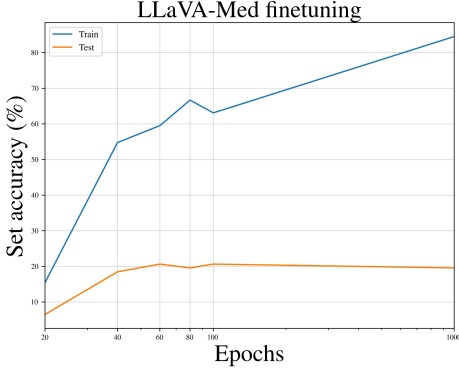

Figure 6: Evolution of train and test set accuracies during LLaVA-Med fine-tuning over epochs.

# E  ABLATION STUDY ON DATASET DIFFICULTY

We investigate the effect of the difficulty of confusing pairs in our benchmark construction. Specifically, we create an 'easy' version of MediConfusion from image pairs that look dissimilar to medical vision-language models, which we refer to as *MediConfusion-easy*. We extract pairs from ROCO that have low BiomedCLIP similarity ($sim_{med} \leq 0.4$). As radiologist feedback is prohibitively costly and time-consuming, we build *MediConfusion-easy* by following the same automatic pipeline as we use in curating MediConfusion, but skip radiologist feedback. Thus, *MediConfusion-easy* may include questions that are clinically irrelevant or have incorrect answers. The resulting dataset consists of 988 pairs (1976 VQA problems).

To make sure that the performance gap between MediConfusion and MediConfusion-easy is not due to the lack of radiologist feedback, we also create *MediConfusion-unfiltered* using identical hyperparameters and procedure to the original benchmark curation, but without radiologist feedback. We match the number of samples to *MediConfusion-easy*.

We evaluate GPT-4o (MC evaluation) and LLaVA-Med (PS evaluation) on MediConfusion-easy. Table 6 summarizes our results. First, we highlight that removing radiologist feedback does not necessarily improve performance (GPT-4o has slightly lower set accuracy, LLaVA-Med performed somewhat better on unfiltered data). Furthermore, our experiment demonstrates that models can indeed differentiate between image pairs in our confusing pair-based VQA format, if the images are different enough, as evidenced by the nearly 100% accuracy of GPT-4o on MediConfusion-easy. Thus, this experiment further supports the need for a strict selection criteria for confusing pairs, as easier variants of our VQA may be unable to identify the shortcomings of state-of-the-art models.

Table 6: Set accuracy, individual accuracy, and confusion (all in %) of GPT-4o (MC evaluation) and LLaVA-Med (PS evaluation) on the easy variant of our benchmark. MediConfusion-easy is not verified by a radiologist, and thus for fair comparison we include an unfiltered version of MediConfusion as well.

| Model | MediConfusion-Easy | | | MediConfusion-Unfiltered | | | MediConfusion | | |
|---|---|---|---|---|---|---|---|---|---|
| | Set acc. | Confusion | Indiv. acc. | Set acc. | Confusion | Indiv. acc. | Set acc. | Confusion | Indiv. acc. |
| GPT-4o | 98.38 | 1.32 | 99.04 | 18.01 | 77.83 | 55.97 | 18.75 | 75.00 | 56.25 |
| LLaVA-Med | 34.21 | 65.59 | 66.90 | 4.86 | 95.75 | 51.06 | 0.00 | 97.16 | 49.72 |

We believe that selecting challenging pairs with high BiomedCLIP similarity, as done in our work, contributes to the difficulty in 2 ways. First, due to the ambiguity of input vision features, the model is unable to resolve enough details in the medical images that is sufficient to answer the questions correctly. Second, the questions themselves are also more challenging, as the difference between hard confusing image pairs is more nuanced. Thus, the model needs more medical knowledge to correctly answer the question, even if the vision embeddings are sufficiently dissimilar. We hypothesize that this medical knowledge is lacking in the case of the Gemini model in our experiments, where we observe poor performance despite low confusion. To highlight how questions are more general and potentially require less medical knowledge, we show an example from MediConfusion-easy in 7, where identifying the anatomical region is sufficient for solving the VQA pair.

# F  INVESTIGATING FEATURE SIMILARITY OF VISION ENCODERS

We find confusing pairs based on BiomedCLIP similarities to construct MediConfusion. The resulting poor performance across all models implies that they struggle differentiating between the confusing pairs, however it is unclear how similar the images are in the respective models' own image feature space. In this section, we analyze the similarity of MediConfusion pairs in the feature space of vision encoders used by open-source models in our experiments: CLIP and the visual encoder of RadFM. RadFM has its own vision encoder trained on medical images, including 3D radiology images. Figure 8 shows the histogram of similarities between confusing pairs based on the cosine similarity of BiomedCLIP, CLIP, and RadFM visual embeddings. We average pool across the token dimension in the output of vision encoders to produce the visual embedding.

As these plots show, the images have high similarity in not only BiomedCLIP's, but other vision encoders' embedding spaces, both trained on general-domain data and medical data.

Question: Is the image showing a condition related to the
respiratory system or the reproductive system?
A: Respiratory system
B: Reproductive system

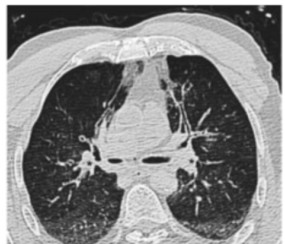 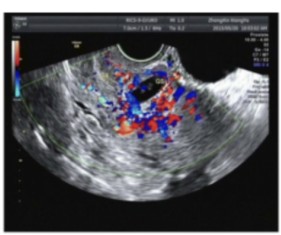

Answer: A                                Answer: B

Figure 7: A sample VQA pair from MediConfusion-easy. Identifying the anatomical region is sufficient to answer the pair correctly.

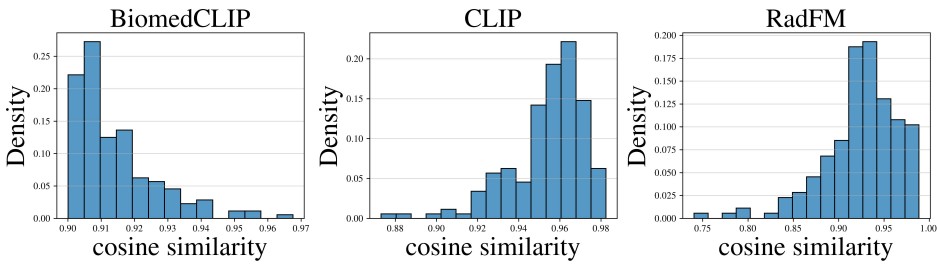

Figure 8: Histograms of the cosine similarity of confusing pairs in different visual representations.

## G    REVERSING THE ROLE OF DINO AND BIOMEDCLIP

As an ablation study, we explore samples that have high DINO similarity but low BiomedCLIP similarity. In particular, we searched for image pairs for which $sim_{med} \leq 0.5$ and $sim_{gen} \geq 0.5$. Figure 9 shows some of these samples.

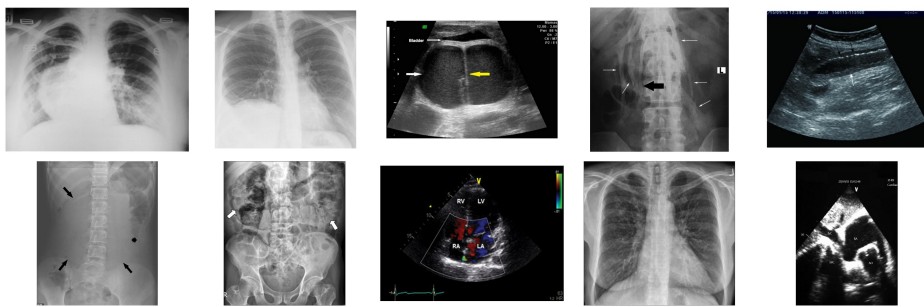

Figure 9: Sample with high DINO similarity but low BiomedCLIP similarity

## H    EXPLORING TECHNIQUES TO IMPROVE PERFORMANCE ON MEDICONFUSION

In this section, we examine various methods we employed to improve model performance on MediConfusion and analyze their effectiveness.

Table 7: Performance of LLaVA-Med model with added DINOv2 embeddings

|  | Set acc. (%) | | | Indiv. acc.(%) | | | Confusion (%) | | | Best | |
|---|---|---|---|---|---|---|---|---|---|---|---|
| Method | MC | GD | PS | MC | GD | PS | MC | GD | PS | Set acc. | Indiv. acc. |
| LLaVA-Med | 0.00 | 0.00 | 1.14 | 23.58 | 49.72 | 49.72 | 100.00 | 99.43 | 97.16 | 1.14 | 49.72 |
| LLaVA-Med + DINOv2 embed. | 2.84 | 2.84 | 2.27 | 45.45 | 50.57 | 50.00 | 95.54 | 95.45 | 95.45 | 2.84 | 50.57 |

Table 8: GPT-4o Best of 100 performance with an oracle as the scoring metric

| Frequency threshold (%) | Set acc. (%) | Indiv. acc. (%) |
|---|---|---|
| 1 | 47.16 | 68.18 |
| 2 | 37.50 | 62.50 |
| 3 | 32.95 | 59.38 |
| 4 | 27.84 | 55.97 |
| 5 | 24.43 | 53.41 |

### H.1 MIXTURE OF FEATURES

Following (Tong et al., 2024b), we investigate the effect of incorporating DINOv2 embeddings into the image encoding of LLaVA-Med. Specifically, we concatenate the DINOv2 embeddings with the existing CLIP embeddings to create an enriched image representation. We then apply the feature alignment and instruction tuning procedures outlined in (Liu et al., 2023) and train the model from scratch. Additionally, we follow the medical-specific feature alignment and instruction tuning methodologies from (Li et al., 2024).

Table 7 compares the performance of this modified model with the original LLaVA-Med. While we observe a slight improvement in accuracy, overall performance remains poor, indicating that richer visual features alone are insufficient to solve MediConfusion.

### H.2 BEST-OF-$N$

The best-of-$N$ method is a widely used approach for enhancing MLLM responses((Stiennon et al., 2020), (Gao et al., 2023), (Yang et al., 2024)). In this technique, we generate $N$ different responses and select the best one based on a predefined scoring metric. For this study, we use GPT-4o and explore different scoring strategies.

#### H.2.1 ORACLE-BASED SCORING

To establish an upper bound on model performance, we employ an oracle-based scoring method, where the oracle has access to the ground truth solutions. We hypothesize that if the model is capable of correctly answering the questions with some reasonable, albeit low, probability, then improved prompting strategies can potentially elicit its medical knowledge. Specifically, we follow the FF evaluation framework, where we first present only the question to the model and generate a response. We then provide GPT-4o with the question, answer choices, and the generated response, asking it to determine which option is closest to the generated answer. We sample $N = 100$ responses for each question. Furthermore, we lower the score gap threshold to $k = 2$. We then measure the frequency with which the correct response appears among these generations. We accept an answer if the frequency of the correct answer exceeds a certain threshold. Table 8 presents the results of this experiment. As we can see, even under these lenient conditions, the model performs poorly, with its performance declining significantly as the acceptance threshold increases.

#### H.2.2 SELF-RANKING

Now, we consider a more realistic setup in which we do not have access to the ground truth. Similar to the previous method, we generate $N$ different samples, then we show the question, options, and all of the generated answers to GPT-4o and ask it to select the best response. We then pass the selected answer as the final answer of the model. Table 9 provides the results of this section for different choices of $N$. Note that since we are using the FF evaluation, in which the model cannot see the

Table 9: GPT-4o best-of-$N$ performance with a judge as the scoring metric

| $N$ | Set acc. (%) | Indiv. acc. (%) | Confusion (%) | Invalid (%) |
|---|---|---|---|---|
| 10 | 5.11 | 33.24 | 81.58 | 41.48 |
| 20 | 4.55 | 30.97 | 81.33 | 43.75 |
| 50 | 7.39 | 36.08 | 79.76 | 38.64 |

Table 10: GPT-4o Majority voting with $N$ answers

| $N$ | Set acc. (%) | Indiv. acc. (%) | Confusion (%) |
|---|---|---|---|
| 1 | 18.75 | 56.25 | 75.00 |
| 10 | 23.30 | 57.39 | 68.00 |
| 100 | 22.73 | 57.39 | 69.32 |

answer options prior to generating the response, the performance is much lower compared to the MC evaluation results in Table 1. These results show that even when we are aggregating 50 responses, the model's performance remains poor, suggesting the presence of fundamental limitations that prevent it from effectively solving these problems.

### H.2.3 MAJORITY VOTING

Self-consistency Wang et al. (2022) sampling can improve reasoning performance by marginalizing over multiple reasoning paths to aggregate the final answer. We generate $N$ different answers using the MC evaluation and then choose the most frequent one as the model's final answer. As shown in Table 10, with $N = 10$ we observe $5\%$ improvement, but increasing $N$ further yields negligible gains.

