# OpenReview forum: "MediConfusion: Can you trust your AI radiologist? Probing the reliability of multimodal medical foundation models"
_ICLR.cc/2025/Conference — ICLR 2025 Poster_

### Official Review · Reviewer_772B · 2024-11-03

**Soundness:** 3
**Presentation:** 3
**Contribution:** 4
**Rating:** 6
**Confidence:** 3

**Summary:**

The paper introduces MediConfusion, a challenging medical Visual Question Answering (VQA) benchmark that exposes the vulnerabilities of current medical multimodal large language models (MLLMs) by highlighting their failure modes. Despite promising progress, existing models—open-source and proprietary—struggle significantly, even performing below random guessing in certain tests. This suggests major reliability issues in deploying these models for healthcare. The benchmark was guided by clinician experts, using adversarial examples to stress-test models, offering a valuable resource for the research community to design more trustworthy MLLMs. Personally I like this paper because it hits on a crucial gap in medical AI that’s often overlooked.

**Strengths:**

1. The paper brings a novel perspective by focusing on the systematic failure modes of medical multimodal models—something that hasn’t been extensively explored before. The fact that state-of-the-art models performed at or below random guessing levels is a striking finding. I can see this dataset becoming a benchmark everyone wants to use as soon as the paper is out, as it offers a strong challenge to prove model reliability.

2. The presentation and writing are clear and well-structured.

**Weaknesses:**

1. One significant limitation is that it feels almost obvious to train a model using both public data and this new dataset to demonstrate improved performance over current models. It’s surprising this wasn’t attempted—even for a dataset paper. Adding such experiments could have strengthened the impact by showing how incorporating MediConfusion directly improves model reliability.

2. The authors might also want to discuss the importance of the image encoder in these models. For example, both LLaVA and LLaVA-med use encoders trained on everyday images, which could be a reason why LLaVA-med performs even worse than LLaVA. If the encoder isn’t well-suited for medical images, it limits the model's ability to learn effectively. Highlighting the role of specialized image encoders could have added valuable insights into improving model reliability in the medical domain.

**Questions:**

See the weakness.

---

> ### Author Response · Authors · 2024-11-22
> **Response to Reviewer 772B**
>
> Thank you for reviewing our paper and for the thoughtful comments. We are glad to hear that the reviewer likes MediConfusion and thinks it “hits on a crucial gap in medical AI that’s often overlooked” and that our paper “brings a novel perspective by focusing on systematic failure modes of medical multimodal models”. Please find below our response to the reviewer's concerns.
>
> **Re weakness 1:** This is a great suggestion. As we summarized in the top-level comment (and Appendix D), we applied the reviewer’s idea and fine-tuned LLaVA-Med on MediConfusion. The detailed findings are in Appendix D, but we give an overview of the results here. We follow standard procedure to fine-tune LLaVA-Med, consisting of fine-tuning the language model weights and the multimodal adapter, while the image encoder is frozen.
>
> Our results show that even after 1000 epochs, the model cannot achieve 100% set accuracy on the training data, indicating that the inputs of some pairs are too similar for the model to distinguish between them. Note, that typical fine-tuning consists of between 1-50 epochs to avoid overfitting. Furthermore, the best test accuracy is around 20%, which is still well below random guessing. These results show that the model likely learned the multiple-choice format due to fine-tuning (improved accuracy from 0% to 20% in multiple-choice evaluation mode), but otherwise the knowledge learned on the train set does not generalize. This strongly implies that vision embeddings of existing models are unable to capture details fine-grained enough for medical reasoning. Thus, in order to improve model robustness and reliability, and to eventually attain a near-perfect score on MediConfusion, we need to improve the vision encoder through improved training data quality (more detailed captions, better alignment, higher quality images), pretraining strategies tailored to the medical domain, or architectural innovations, all of which are exciting directions for future work.
>
> **Re weakness 2:** Thank you for this insightful suggestion. We agree that the visual encoder plays an important role, and the lack of a specialized vision encoder can limit the models. Also, the models need to have the medical knowledge necessary to answer these questions. However, it is worth noting that RadFM has its own vision encoder trained on medical images, including 3D radiology images, but it still fails on MediConfusion.  Furthermore, following the reviewer’s suggestion, we perform additional experiments on the vision features of common models. The detailed results are included in Appendix F, and we have a summary in the top-level comment (point 4).
>
> In particular, we analyze the similarity of MediConfusion pairs in the feature space of vision
> encoders used by open-source models in our experiments: CLIP and the visual encoder of RadFM. We average pool across the token dimension in the output of vision encoders to produce the visual embedding. The histograms of confusing pair similarities depicted in Figure 8 in our updated manuscript show that the image pairs have high similarity in not only BiomedCLIP’s, but other vision encoders’ embedding spaces, both trained on general-domain data and medical data. We would also like to emphasize that producing different enough embeddings for confusing pairs is a necessary but not sufficient condition to tackle MediConfusion. Specifically, Gemini tends to select different answers for questions within a pair (implying it is able to differentiate between them),  however, its overall performance is still below random guessing, highlighting the importance of medical knowledge as well.

---

> > ### Author Response · Authors · 2024-11-25
> > **Asking Reviewer 772B for feedback on the updated manuscript**
> >
> > We are grateful for the reviewer’s time and effort in reviewing our paper. We also thank the reviewer for the great suggestions, we added a new experiment on fine-tuning on MediConfusion, and one on visual feature similarities across different models. We hope that our rebuttal answered all the reviewer’s questions and concerns with respect to our paper. As the discussion period is closing soon, we kindly ask the reviewer to let us know if there are any further questions about our paper. We would greatly appreciate it if the reviewer could raise their rating if satisfied with our rebuttal, given the overall favorable review and high scores. Thank you again for your invaluable feedback.

---

### Official Review · Reviewer_cT2V · 2024-11-04

**Soundness:** 3
**Presentation:** 3
**Contribution:** 2
**Rating:** 5
**Confidence:** 4

**Summary:**

This paper evaluates the reliability of medical Multimodal Large Language Models in healthcare. The authors present MediConfusion, a new VQA benchmark that highlights distinguishing similar medical xray images. They find that all tested MLLMs, open-source and proprietary, perform below random chance on MediConfusion, indicating serious reliability issues.

**Strengths:**

- The idea of using the same question with similar images could offer a interesting way to probe model weaknesses.

- With the help of an expert-in-the-loop pipeline, the paper thoroughly examines why models struggle to differentiate between visually similar medical images.

- The evaluation metrics are comprehensive, particularly accommodating models unable to directly answer multiple-choice questions.

**Weaknesses:**

- The motivation is not clear. The authors claimed that "existing benchmark datasets are focused on evaluating the medical knowledge of MLLMs across large evaluation sets, heavily biased towards common or typical scenarios." What are the common and typical scenarios? Could you provide a more detailed discussion about this limitation of existing benchmarks and how the proposed benchmark in this paper can overcome the limitations?
- The results show that all available models (open-source or proprietary) achieve performance below random guessing on MediConfusion. It would be better if the authors conducted more experiments to find the reasons. For example, by using different sim_med and sim_gen thresholds and compare the model performance, we can identify whether the models can differentiate easy image pairs or they even cannot differentiate any image pairs.
- The writing style needs refinement. For instance, the phrase "stress-tests the visual capabilities of state-of-the-art models" is unclear and may confuse readers. The motivation and innovative aspects of the benchmark remain vague until "image pair" is mentioned at the end of Section 2, and even then, it is not clarified that the task is framed as multiple-choice questions. It would be easier for the readers to better understand this paper if the writer could clarify these designs in the introduction section.
- The rationale for focusing on the medical domain is unclear. The methods do not appear specific to this field, resembling a rephrasing of existing VQA datasets where answers are scrambled, and models are tasked with similar image comparisons. Could you explain what is the specific design of this benchmark considering its medical context? What is the difference between two similar medical images and two similar general images?
- The choice to format the task as multiple-choice is questionable. Have you considered open-ended QA or report generation of CXR images? Additionally, given the relatively small dataset (352 questions), this benchmark may lack the scale needed to effectively assess MLLM capabilities. Could you justify that 352 questions are sufficient for the evaluation?

**Questions:**

- Does the proposed benchmark dataset only contain 352 image pairs?

- The authors only evaluate the models on the similar pair images. Can the models work well on different image pairs? For example, to use different sim_med and sim_gen thresholds and compare the model performance.

---

> ### Author Response · Authors · 2024-11-22
> **Response to Reviewer cT2V (1)**
>
> We thank the reviewer for their invaluable insight. We are glad that the reviewer finds that our work “thoroughly examines why models struggle to differentiate between visually similar medical images” and that the evaluation metrics are “comprehensive”. Please find below our response to the reviewer's concerns.
>
> **Re weakness 1:** Great question. Most medical benchmarks for evaluating MLLMs test models on general medical VQA, where the specific shortcomings of visual representations and limitations to discern nuanced but medically relevant details (the scenarios that our work is targetting) may be outnumbered by simple questions that can be easily solved by more superficial medical image understanding capabilities, and in some cases even without looking at the image. For instance, LLaMa, which is a language-only model, achieves non-trivial accuracy on PMC-VQA (as it was reported by their paper) without seeing the images. This may result in high reported accuracies in such benchmarks, as the failure cases are averaged into a large number of standard (and potentially easy to solve) problems. However, in a safety-critical setting, such as healthcare, it is of utmost importance to understand the limitations of AI solutions. To overcome this limitation of other benchmarks, we specifically design MediConfusion to probe a particular type of failure mode originating in the ambiguity of visual embeddings, resulting in a challenging benchmark that forces the model to “look at” the image (language-only accuracy is random guessing by design). We hope that this discussion clarifies our motivation.
>
> **Re weakness 2:** Thank you for raising this great question. One obstacle to simply varying the benchmark generation thresholds is that each final pair included in MediConfusion, takes 15 minutes of radiologist time on average, including the amortized time to filter through a large pool of candidates, review the pair for correctness and relevance and to improve the language if needed.  Overall, it takes around 40 hours of direct radiologist involvement to generate a dataset with approximately 350 samples. As a result, it is extremely costly to perform extensive ablations on dataset curation parameters. That said, we find the reviewer’s question very relevant and interesting, therefore we create an “easy” dataset while skipping radiologist feedback on the included pairs, which we call *MediConfusion-easy*. We add details in Appendix E. MediConfusion-easy consists of approx. 1000 pairs with BiomedCLIP similarity below 0.4, indicating that they should be clearly distinct for medical models. We tested GPT-4o and LLaVA-Med on MediConfusion-easy, and the results are below.
>
> | Model        | MediConfusion-Easy | Unfiltered MediConfusion | MediConfusion |
> |--------------|---------------------|--------------------------|---------------|
> | GPT-4o       | 98.38              | 18.01                   | 18.75         |
> | LLaVA-Med    | 33.91              | 5.87                    | 0.0           |
>
> Our experiment demonstrates that models can indeed differentiate between image pairs in our confusing pair-based VQA format, if the images are different enough, as evidenced by the nearly 100% accuracy of GPT-4o on MediConfusion-easy and the significant improvement in LLaVA-Med’s set accuracy. Thus, this experiment further supports the need for a strict selection criteria for challenging confusing pairs, as easier variants of our VQA may be unable to identify the shortcomings of state-of-the-art models. To ablate the effect of removing radiologist feedback from dataset curation, we create an unfiltered MediConfusion variant using an identical methodology to MediConfusion but skipping radiologist filtering. As shown in the table, removing radiologist feedback does not necessarily improve performance (GPT-4o has slightly lower set accuracy, and LLaVA-Med performed somewhat better on unfiltered data).
>
> **Re weakness 3:** We thank the reviewer for pointing this out. We added more clarification to this discussion to remove any ambiguity. We highlighted the relevant section in the Introduction of the updated manuscript. We hope that the reviewer finds the language to be more precise in the updated version.

---

> > ### Author Response · Authors · 2024-11-22
> > **Response to Reviewer cT2V (2)**
> >
> > **Re weakness 4:** The key rationale for introducing MediConfusion is the importance of understanding the limitations and reliability of MLLMs in the medical domain, due to the safety-critical nature of healthcare applications. Our benchmark construction process is very specific to the medical domain, as we use captioned radiology data to build the dataset, we base our selection criteria on biomedical vision-language models and we crucially rely on medical experts to verify and rephrase VQAs to reflect clinically relevant problems. Moreover, we extract failure modes specific to the field of radiology that cannot be performed via general domain methods. Our focus is not necessarily to invent a new VQA format for the medical domain, but to provide the community with a challenging  (even for highly performant, state-of-the-art proprietary models) medical benchmark designed to test for shortcomings in medical image understanding and reasoning originating in ambiguities in visual embeddings.
> >
> > **Re weakness 5:** Thank you for raising this point. We argue that it is much more straightforward and well-defined to evaluate models given a closed set of options (multiple-choice format). Despite limiting the answer options, we ensure that our evaluation is fair, as we include free-form answer generation with GPT scoring and prefix scoring as well in our evaluations. Before deciding on the format we also consulted with top experts both in industry and academia all of whom suggested that it is best for an evaluation benchmark to be multiple-choice (to avoid the challenges/ambiguities above). That being said, the reviewer highlighted an exciting direction toward probing failure modes in medical report generation, a task of much higher complexity than simple multiple-choice VQA (both in terms of model outputs and performance evaluation). We believe that addressing the shortcomings revealed in a simpler setting by MediConfusion is a necessary first step towards deploying these models on more complex, and clinically highly relevant applications such as radiology report generation.
> >
> > **Re question 1:** MediConfusion consists of 352 VQA problems. As we have highlighted for another reviewer,  MediConfusion is an ongoing project with further additions to come in the near future.  However, due to the need for rigorous radiologist review, the curation process is expensive and time-consuming. We have a large dataset of automatically generated pairs that have not been reviewed by radiologists, which can be useful for medical MLLM training in future work. That said, we believe that our benchmark clearly demonstrates a fundamental shortcoming of existing models, as all of the models consistently perform below random guessing.
> >
> > **Re question 2:** Please see our detailed answer to this question in response to Weakness 2.

---

> > > ### Author Response · Authors · 2024-11-25
> > > **Asking Reviewer cT2V for feedback on the updated manuscript**
> > >
> > > We appreciate the reviewer’s comments on our paper, we did our best to incorporate all the feedback and further strengthen our work. We added a new experiment following the reviewer’s exciting idea of verifying confusing pairs in an “easier” setting, and fixed the confusing language in the introduction. We hope that the reviewer finds our rebuttal and the additional updates to the manuscript satisfactory. As the end of the rebuttal period is closing in, we would like to ask whether the reviewer has any lingering concerns or questions with respect to our manuscript. We would appreciate it if the reviewer could raise their rating if we have managed to address their concerns. Thank you again for the valuable feedback.

---

> > ### Comment · Reviewer_cT2V · 2024-11-25
> >
> > Thanks for your reply.
> >
> > For the MediConfusion-easy dataset, are each pair of images the same image type (e.g., CT of the brain, or XRay of the lung)?
> > I am not sure whether the LLMs can capture any anomaly features.
> >
> > For the new results on MediConfusion-Easy, LLaVA-Med still significantly performs worse than random guessing. Does it mean LLaVA-Med learns nothing from the images?

---

> ### Author Response · Authors · 2024-11-25
> **Response to  Reviewer cT2V**
>
> Thanks for getting back to us.
>
> Re question on modality:
> We followed an identical procedure when curating the easy variant as the original benchmark, and therefore we didn’t apply any hard constraints on the modality of image pairs. That said, due to the relaxed conditions on confusing pairs, the images in the pair are often of different modality or anatomical region. Please find an example image pair in Appendix E (Figure 7.) and a more detailed discussion on MediConfusion-easy. We believe that in this relaxed setting, the MLLMs can leverage less nuanced medical features to differentiate between the images, thus the improved accuracy.
>
> Re question on accuracies:
> For MediConfusion-Easy, the 34% accuracy reported for LLaVA-Med is “set accuracy” which is 25% for random guessing (only counts as “hit” if both answers are correct within a pair of 2-choice questions). The standard notion of (individual) accuracy of this model on MediConfusion-Easy is 66.9%, which is also higher than random guessing (50%). We believe that both LLaVA-Med’s image encodings (confusion is still 65.6%) and medical knowledge are lacking the necessary detail to achieve better performance on this dataset. We have evidence that the easy benchmark can be solved to near-perfect accuracy, as GPT-4o achieves 98.38% set accuracy (99.04% individual accuracy) without any fine-tuning. We have added individual accuracy and confusion along with the set accuracy in the corresponding table in the updated manuscript to avoid future misunderstanding.
>
> Please let us know if you have any further questions.

---

> > ### Comment · Reviewer_cT2V · 2024-11-26
> >
> > Thank you for your reply. I don't have additional questions and updated the scores.

---

### Official Review · Reviewer_bwv6 · 2024-11-04

**Soundness:** 4
**Presentation:** 4
**Contribution:** 4
**Rating:** 8
**Confidence:** 3

**Summary:**

This paper evaluates reliability of medical VLMs. The authors introduce MediConfusion, a benchmark comprising 352 pairs of “confusing” medical images. Each pair consists of images that are visually distinct yet similar in feature space. The authors leverage GPT-4 to generate questions for which the answers differ between the two images. The benchmark data is further evaluated by a radiologist. Experimental results indicate that MediConfusion poses significant challenges to several existing VLMs, highlighting areas for improvement and setting the stage for future research.

**Strengths:**

1. The paper addresses a well-motivated and underexplored problem in the medical domain.
2. The benchmark creation process is robust, with manual evaluation by a radiologist enhancing its credibility.
3. The experiments are comprehensive, demonstrating that MediConfusion presents substantial challenges to current VLMs, with promising potential for inspiring future research.

**Weaknesses:**

While this work provides valuable insights, it appears to closely follow the Multimodal Visual Patterns (MMVP) benchmark framework, potentially limiting its novelty. Many findings align with known challenges in general VLMs.

**Questions:**

1. Are there confusing pairs that appear visually similar but differ in feature space?
2. How would the Mixture-of-Features method from "Eyes Wide Shut? Exploring the Visual Shortcomings of Multimodal LLMs" apply to medical VLMs?
3. What are the limitations of the confusing pairs identified here? Could there be potential biases?

---

> ### Author Response · Authors · 2024-11-22
> **Response to Reviewer bwv6**
>
> We thank the reviewer for their effort in reviewing our work. We are glad that the reviewer finds the problem we target is “well-motivated and underexplored”, that our “benchmark creation process is robust” and believes that our “experiments are comprehensive”. Please find below our response to the concerns and questions.
>
> **Re weakness:** Thank you for raising this point. Our novelty originates in focusing on failure modes in the medical domain, where bringing attention to limitations and shortcomings is crucial for safe deployment. Before our work, it had been unclear whether general domain failure modes even appear in medical MLLMs, due to significant differences in the data distribution. Specifically, most ambiguities in general-domain vision-language models, such as CLIP, have been traced back to poor pre-training data quality: images are typically not similar enough within a training batch to force the model to learn fine-grained representations in order to differentiate between them and thus the model can rely on “shortcuts” to minimize the contrastive loss. It is unclear if the same phenomenon appears when pre-training medical MLLMs, as both the image and caption distribution are vastly different from general-domain data. Our work reveals the novel insight that in fact medical MLLMs are affected by similar shortcomings, and therefore we provide a challenging benchmark to the community to assess and eventually improve the reliability of MLLMs in the healthcare domain.
>
> **Re question 1:** This is a very interesting question. We find such pairs by searching for images with high DINO similarity, but low BiomedCLIP similarity. We included sample pairs in Appendix G in Figure 9. As the reviewer can see, these pairs have similar overall structures (e.g., sonography scans) that can be confusing for a general domain purely vision model, such as DINO, but a medical vision encoder can distinguish between them. We note that in MediConfusion, all pairs have low DINO similarities and high BiomedCLIP similarities, which is the opposite of the condition we used to find these specific samples.
>
> **Re question 2:** Thank you for this great suggestion. Our new experiments (please see top comment 3 for an overview and Appendix D for details) demonstrate that after fine-tuning the language model and multimodal adapter of LLaVA-Med on MediConfusion, the model cannot achieve perfect score on the training data even after a very high number of fine-tuning epochs. This indicates that the frozen visual embeddings are not detailed enough to distinguish between images in some confusing pairs. Novel approaches to improve the vision inputs of medical MLLMs are required to improve performance on MediConfusion, and thus improve their reliability in healthcare. We believe that improved training data quality (more detailed captions, better alignment, higher quality images) and pre-training strategies, architectural innovations, and potentially combining vision features from various models, as the reviewer has suggested,  can potentially address the shortcomings, which are all interesting directions for future work.
>
> **Re question 3:** Thank you for raising this point, this is an important aspect of benchmarking. As depicted in Figure 4 in our paper, some categories, like vascular images, have higher representation in MediConfusion. This is due to their more frequent occurrence in medical datasets in general,  (due to their importance in clinical practice) resulting in an overall higher number of valid VQAs after radiologist filtering.  We expect models to perform better in such categories (as they have likely seen more samples of these in their training set), and indeed, this is what we observe with GPT-4o (see categorical breakdown of results in Table 2).

---

> > ### Author Response · Authors · 2024-11-25
> > **Asking Reviewer bwv6 for feedback on the updated manuscript**
> >
> > We are grateful for the reviewer’s insights and feedback on our work. As the discussion period is closing soon, we would like to ask if the reviewer has any further unanswered questions with respect to our paper. Thank you again for reviewing our manuscript.

---

### Official Review · Reviewer_QM8Q · 2024-11-04

**Soundness:** 3
**Presentation:** 3
**Contribution:** 3
**Rating:** 6
**Confidence:** 4

**Summary:**

The paper introduces MediConfusion, a benchmark specifically designed to assess and expose failure modes in multimodal large language models (MLLMs) used in medical visual question answering (VQA) tasks. The benchmark highlights that current state-of-the-art MLLMs, including both open-source and proprietary models, struggle significantly with distinguishing visually dissimilar yet clinically distinct medical images, often performing below random guessing on the dataset. MediConfusion is created by identifying “confusing image pairs” from the ROCO radiology dataset, where image pairs appear distinct to medical professionals but prove challenging for MLLMs due to limitations in visual feature space alignment.

The authors reveal key failure patterns among MLLMs, such as difficulties in differentiating between normal and pathological anatomy, recognizing lesion characteristics, and identifying medical devices and vascular conditions. The benchmark leverages radiologist oversight to ensure clinical relevance and accuracy in the question-answer pairs, aiming to guide future MLLM improvements in medical contexts. By exposing these systematic vulnerabilities, the paper underscores the critical need for more reliable and trustworthy AI systems in healthcare applications.

**Strengths:**

Originality: The paper addresses a critical gap in the assessment of medical MLLMs by introducing MediConfusion, a novel benchmark explicitly designed to test the reliability of these models in healthcare. While prior benchmarks focus on overall performance in typical medical scenarios, this work innovatively emphasizes systematic failure modes by identifying “confusing image pairs” that challenge models with subtle, clinically important distinctions. The benchmark not only probes weaknesses but also highlights specific visual reasoning failures, which is a unique contribution to medical AI evaluation.

Quality: The dataset creation is rigorous, combining automated techniques with expert radiologist validation to ensure that each question is clinically relevant and accurately represents medical knowledge. The experimental design is robust, systematically evaluating a range of state-of-the-art models (both general and medical-domain MLLMs) and comparing their performance against baseline expectations. The detailed breakdown of common model failure patterns enhances the paper’s depth, providing actionable insights for improving MLLM robustness in medical settings.

Clarity: The paper is well-structured and presents its findings in a clear, logical manner. Descriptions of the dataset creation process, evaluation methods, and results are easy to follow, with visual aids (e.g., figures illustrating confusing image pairs) that help readers understand the complexity of the visual tasks posed to models. The breakdown of error types further clarifies the distinct challenges MLLMs face, enhancing the reader’s comprehension of the benchmark’s significance.

Significance: The benchmark’s focus on failure modes has broad implications for the deployment of MLLMs in healthcare. By surfacing limitations that could compromise patient safety, the work serves as a foundational tool for future research aimed at increasing AI reliability in clinical practice. Additionally, MediConfusion highlights the limitations of current visual encoders and training paradigms in handling medical data, offering a clear research direction for building more robust and trustworthy models. The benchmark’s significance is further bolstered by its potential as a standard for evaluating and guiding improvements in medical AI systems.

**Weaknesses:**

Limited Dataset Size: While the MediConfusion benchmark is rigorous, the dataset size (352 questions across 9 categories) may be relatively small to capture the full spectrum of medical image complexity. Expanding the dataset with a broader range of confusing image pairs, potentially covering additional anatomic regions and diagnostic subtleties, could improve its generalizability and make it more comprehensive for model evaluation across diverse medical contexts.

**Questions:**

1.	Dataset Diversity and Expansion: Could you elaborate on the decision to limit the benchmark to the ROCO dataset? Do you have plans to expand MediConfusion to include other datasets, such as ultrasound or nuclear imaging, to capture a broader range of diagnostic tasks? Additional dataset diversity might reveal new model weaknesses or challenges in different medical imaging modalities.
	2.	Impact of Prompt Engineering: Given the importance of prompt phrasing in VQA tasks, did you experiment with alternative prompt formats or styles (e.g., framing questions in different ways or including detailed instructions)? If so, what impact did this have on model performance, and could further prompt optimization improve reliability? Insights here could guide future research on prompt refinement in medical VQA.
	3.	Future Directions for Medical MLLM Improvements: Given the identified failure patterns (e.g., difficulties with spatial reasoning or lesion characteristics), do you have specific recommendations for architectural or training adjustments that could address these weaknesses? For example, do you see value in integrating spatial attention mechanisms or training models on more anatomically detailed data?

---

> ### Author Response · Authors · 2024-11-22
> **Response to Reviewer QM8Q**
>
> We thank the reviewer for their valuable feedback on our manuscript. We are glad to hear that the reviewer finds the paper to be a “novel” and “unique contribution to medical AI evaluation”, and that our work has “broad implications on the deployment of MLLMs in healthcare. Please find below our response.
>
> **Re weakness:** Thank you for this great suggestion. MediConfusion is an ongoing project with further additions and expansions yet to come with more categories and modalities.  However, due to the need for rigorous radiologist review, the curation process is expensive and time-consuming. We have many more automatically generated samples that have not been reviewed by radiologists yet, which can be useful for training medical MLLMs in future work.
>
> **Re question 1:** Thank you for raising this point. Our primary focus so far for MediConfusion is on radiology due to its enormous clinical impact. The ROCO dataset has a very high number of radiology images with detailed captions, making it a perfect fit for our cause. We plan to expand Mediconfusion with more pairs and add more samples from tasks other than radiology. MediConfusion is our first step towards a collection of challenging medical benchmarks covering diverse modalities.
>
> **Re question 2:** This is a good suggestion. In our experiments,  we set up the prompt following recommendations for the specific model as closely as possible. Following the reviewer’s suggestion, we performed additional ablation studies on the effect of prompting on MediConfusion performance. We recap our experiments from the top-level comment here. We create 10 variants of the multiple-choice input prompt and record the performance of GPT-4o, InstructBLIP, and LLaVA-Med. The table below shows the mean and standard deviation of the performance:
>
> | Model         | Set Score (Mean) | Set Score (Std) |
> |---------------|------------------|-----------------|
> | GPT-4o        | 19.12           | 0.043           |
> | InstructBLIP  | 4.79            | 0.116           |
> | LLaVA-Med     | 0               | 0               |
>
> We observe that GPT-4o performance is robust to prompt format, but the performance is still below random guessing. LLaVA-Med struggles with the multiple-choice format, and using different prompts does not resolve this issue. For InstructBLIP, the sensitivity to prompting is higher than GPT-4o, but its performance does not change drastically. Overall, we find that prompt format and style are likely not the root cause for poor performance on MediConfusion. For more details on the specific prompts we used please find the experimental details in Appendix C. Thanks a lot again for this great suggestion.
>
> **Re question 3:** This is a very important question. Our new fine-tuning experiment (Appendix D, or please see top-level comment for a summary) indicates that the vision embeddings likely don’t capture enough detail to effectively differentiate between images in confusing pairs, as evidenced by the fact that LLaVA-Med cannot even overfit to MediConfusion when trained on the dataset (vision encoder kept frozen). This leads us to believe that the pretraining data quality (caption density, image-caption alignment, image resolution and diversity) used for medical vision encoder training needs to be greatly improved (as the reviewer has also suggested). We believe that incorporating sets of similar images (“confusing sets”) and corresponding anatomically detailed captions into the pre-training data would have the largest impact on improving the vision encoder. In this scenario, the model would be forced to  learn more refined and distinguishing image representations for medical images, in order to minimize the contrastive pre-training loss. In tandem with improvements in data quality, the vision encoder architecture also has to be improved in its capacity to represent nuanced medical image features in order to fully utilize the improved dataset. This can be done through spatial attention mechanisms, as the reviewer suggested, an increased (or adaptive) number of visual tokens, and incorporating multi-scale image representations. Fundamental architectural innovations for medical MLLMs is an interesting direction for future research.

---

> > ### Author Response · Authors · 2024-11-25
> > **Asking Reviewer QM8Q for feedback on the updated manuscript**
> >
> > We thank the reviewer again for their invaluable feedback on our work, and we hope that the new experiment we added on prompt engineering and our response to the reviewer’s additional questions have addressed the reviewer’s concerns. As the end of the discussion period is getting close, we would like to ask if there are any lingering questions we could clarify. If the reviewer is satisfied with our rebuttal and the new experiment, we would like to ask to consider raising the rating, given the overall favorable review and high scores. Thanks a lot again for the great suggestions.

---

### Author Response · Authors · 2024-11-22
**Summary of additional experiments (1)**

We appreciate the time and effort of all reviewers and their insightful feedback. We did our best to address all concerns regarding our work. We performed additional experiments to reflect reviewer comments and updated the manuscript accordingly (highlighted in blue). The summary of new experiments is as follows:

1. **Prompt sensitivity:** As suggested by **Reviewer QM8Q**, we add an ablation study on the effect of manual prompt engineering on the performance of the models in Appendix C.  For this study, we create 10 variants of the multiple-choice input prompt and record the performance of GPT-4o, InstructBLIP, and LLaVA-Med. The table below shows the mean and standard deviation of the performance:

| Model         | Set Score (Mean) | Set Score (Std) |
|---------------|------------------|-----------------|
| GPT-4o        | 19.12           | 0.043           |
| InstructBLIP  | 4.79            | 0.116           |
| LLaVA-Med     | 0               | 0               |

We observe that GPT-4o performance is robust to prompt format, but the performance is still below random guessing. LLaVA-Med struggles with the multiple-choice format, and using different prompts does not resolve this issue. For InstructBLIP, the sensitivity to prompting is higher than GPT-4o, but its performance does not change drastically. Overall, we find that prompt format and style is likely not the root cause for poor performance on MediConfusion.

2. **Effect of dataset difficulty:** In Appendix E, we add a discussion on the effect of similarity thresholds on dataset difficulty based on **Reviewer cT2V**’s suggestion. We create an ‘easy’ version of the benchmark by specifically looking for image pairs that are dissimilar (cosine similarity below 0.4) in BiomedCLIP’s feature space. More details on dataset creation are included in the appendix.  We find that the performance of GPT-4o and LLaVA-Med increased significantly (to 98.38% and 33.91% set accuracy respectively):

| Model                | Easy Variant | Original MediConfusion |
|----------------------|--------------|------------------------|
| GPT-4o (MC eval.)    | 98.38        | 18.01                 |
| LLaVA-Med (PS eval.) | 34.21        | 1.14                   |

Our experiment demonstrates that models can indeed differentiate between image pairs in our confusing pair-based VQA format, if the images are different enough, as evidenced by the nearly 100% accuracy of GPT-4o on the easy variant. Thus, this experiment further supports the need for a strict selection criteria for confusing pairs, as easier variants of our VQA may be unable to identify the shortcomings of state-of-the-art models.

3. **Effect of fine-tuning:** Following the suggestion of **Reviewer 772B**, we add experiments on the effect of fine-tuning on MediConfusion in Appendix D. We follow standard procedure to fine-tune LLaVA-Med, consisting of fine-tuning the language model weights and the multimodal adapter, while the image encoder is frozen. We train on approx. half of MediConfusion samples and evaluate performance on the other half. The following table summarizes our results:

| Epoch | Train Set Accuracy | Test Set Accuracy |
|-------|---------------------|-------------------|
| 20    | 15.48              | 6.52              |
| 40    | 54.76              | 18.48             |
| 60    | 59.52              | 20.65             |
| 80    | 66.67              | 19.57             |
| 100   | 63.10              | 20.65             |
| 1000  | 84.52              | 19.57             |

Our results show that even after 1000 epochs, the model cannot achieve 100% set accuracy on the training data, indicating that the inputs of some pairs are perceived as being too similar by the model to distinguish between them despite being distinct. Note that typical fine-tuning consists of between 1-50 epochs to avoid overfitting. Furthermore, the best test accuracy is around 20%, which is still well below random guessing. These results show that the model likely learned the multiple-choice format due to fine-tuning (improved accuracy from 0% to 20%), but otherwise the knowledge learned on the train set does not generalize, which strongly implies that vision embeddings are too ambiguous. This experiment suggests that MediConfusion can likely only be solved by improving the vision encoder directly.

---

> ### Author Response · Authors · 2024-11-22
> **Summary of additional experiments (2)**
>
> 4. **Input feature similarity for additional image encoders:** As suggested by **Reviewer 772B**, in Appendix F we performed additional experiments on the vision embeddings of various models.  In particular, we analyze the similarity of MediConfusion pairs in the feature space of vision encoders used by open-source models in our experiments: CLIP and the visual encoder of RadFM. RadFM has its own vision encoder trained on medical images, including 3D radiology images. Our experiments highlight that, even though we constructed our benchmark based on BiomedCLIP features, the confusing pairs have highly similar features in other models’ representation space as well. Interestingly, we don’t find major differences between general-domain (CLIP) and medical vision-language models (BiomedCLIP, RadFM encoder) in this regard, potentially indicating the need for improved data and training practices.

---

### Meta-Review · Area_Chair_bivX · 2024-12-19

**Metareview:**

This is a datasets and benchmarks paper. The authors introduced MediConfusion, a challenging medical Visual Question Answering (VQA) benchmark dataset that probes the failure modes of medical MLLMs from a vision perspective. The benchmark dataset is designed to evaluate the ability of state-of-the-art models to recognize subtle yet clinically meaningful differences between medical images. Then, the dataset was further evaluated by a radiologist. The authors find that all tested MLLMs, open-source and proprietary, perform below random chance on MediConfusion, indicating some serious reliability problems on MLLMs. In sum: this is an interesting and timely paper on hot topics - medical MLLMs, the findings are unexpected - all models from GPT-4o, Gemini 1.5 Pro to Llava, InstructBLIP, achieve performance below random guessing on MediConfusion, and the code and data were provided with the paper. After the rebuttal stage, most reviewers supported this paper.

**Additional Comments On Reviewer Discussion:**

The authors have sufficiently addressed most comments from all the reviewers. Most reviewers have increased their scores accordingly during the discussions.

---

### Decision · Program_Chairs · 2025-01-22

Accept (Poster)